# Bioavailable atmospheric phosphorous supply to the global ocean: a 3-D global modelling study

Stelios Myriokefalitakis[1], Athanasios Nenes[2,3,4,5], Alex R. Baker[6], Nikolaos Mihalopoulos[1,5] and Maria Kanakidou[1]

[1]Environmental Chemical Processes Laboratory, Department of Chemistry, University of Crete, P.O. Box 2208, 70013 Heraklion, Greece
[2]Institute of Chemical Engineering Sciences (ICE-HT), FORTH, P.O. Box 1414, 26504 Patras, Greece
[3]School of Earth and Atmospheric Sciences, Georgia Institute of Technology, 311 Ferst Drive, Atlanta, GA 30332-0100, USA
[4]School of Chemical and Biomolecular Engineering, Georgia Institute of Technology, 311 Ferst Drive, Atlanta, GA 30332-0100, USA
[5]National Observatory of Athens, Institute for Environmental Research and Sustainable Development, Athens
[6]Centre for Ocean and Atmospheric Science, School of Environmental Sciences, University of East Anglia, Norwich UK, NR4 7TJ

*Correspondence to*: Stelios Myriokefalitakis (stelios@uoc.gr); Maria Kanakidou (mariak@uoc.gr)

**Abstract.** The atmospheric cycle of phosphorus (P) is here parameterized in a state-of-the-art global 3-D chemistry-transport model, taking into account primary emissions of total P (TP) and soluble P (DP) associated with mineral dust, combustion particles from natural and anthropogenic sources, bioaerosols, sea-spray and volcanic aerosols. For the present day, global TP emissions are calculated to be roughly 1.33 Tg-P yr$^{-1}$, with the mineral sources contributing more than 80% to these

emissions. The P solubilization from mineral dust under acidic atmospheric conditions is also parameterized in the model and is calculated to contribute about one third (0.14 Tg-P yr$^{-1}$) of the global DP atmospheric source. To our knowledge, a unique aspect of our global study is the explicit modeling of the evolution of phosphorus speciation in the atmosphere. The simulated present day global annual DP deposition flux is 0.45 Tg-P yr$^{-1}$ (about 40% over oceans), showing a strong spatial and temporal variability. Present day simulations of atmospheric P aerosol concentrations and deposition fluxes are

satisfactory compared with available observations, indicating however an underestimate of about 70% on current knowledge of the sources that drive P atmospheric cycle. Sensitivity simulations using preindustrial (year 1850) anthropogenic and biomass burning emission scenarios showed a present-day increase of 75% in the P solubilisation flux from mineral dust, i.e. the rate at which P is converted into soluble forms, compared to preindustrial times, due to increasing atmospheric acidity over the last 150 years. Future reductions in air pollutants, due to the implementation of air-quality regulations, are expected

to decrease the P solubilisation flux from mineral dust by about 30% in the year 2100 compared to the present-day. Considering however that all the P contained in bioaerosols is readily available for uptake by marine organisms, and also accounting for all other DP sources, a total bioavailable P flux of about 0.17 Tg-P yr$^{-1}$ to the oceans is derived. Our calculations further show that in some regions more than half of the bioavailable P deposition flux to the ocean can originate from biological particles, while this contribution is found to maximize in summer when atmospheric deposition impact on

the marine ecosystem is the highest due to ocean stratification. Thus, according to this global study, a largely unknown but potentially important role of terrestrial bioaerosols as suppliers of bioavailable P to the global ocean is also revealed. Overall, this work provides new insights to the atmospheric P cycle by demonstrating that biological material are important carriers of bioavailable P, with very important implications for past and future responses of marine ecosystems to global change.

# 1    Introduction

Phosphorus (P) is a ubiquitous element found in amino-acids, in proteins and as an integral part of organisms, together with nitrogen (N) and iron (Fe). It is an essential nutrient that can limit primary production and nitrogen-fixation in aquatic environments and thus significantly influence carbon-storage (Elser et al., 2007). Reviewing experimental data, Moore et al. (2013) proposed two broad regimes of phytoplankton nutrient limitation in the modern upper ocean: 1) N-limited regimes in most of the low latitude oceanic surface and 2) Fe-limited regimes where subsurface nutrient supply is enhanced; while P may co-limit primary productivity. Moutin et al. (2008) pointed out the potential importance of phosphate for $N_2$ fixation in particular in the Southeast Pacific under high temperature conditions and Fe availability, favourable for the presence of $N_2$-fixing organisms (like *Trichodesmium* spp.) that potentially counteract the N-limitation (Deutsch et al., 2007). However, in some regions like the Mediterranean, primary productivity is found to be limited by P- availability to the marine ecosystems (Krom et al., 2005). Furthermore, (Brahney et al., 2015) and (Du et al., 2016) found that human driven imbalanced atmospheric deposition of N and P might have induced or will induce P-limitation to the ecosystems (global alpine lakes and large areas of China's forests, respectively).

The two external-to-the-ocean sources of nutrients are the atmosphere and rivers. Depending on these inputs and marine dynamics, different nutrients can limit the marine primary productivity. Riverine inputs of nutrients to the marine ecosystem are important for coastal regions, while the atmospheric deposition of nutrients is a more significant source to the open ocean (Jickells, 2005; Duce et al., 2008; Mahowald et al., 2008). In contrast to the atmospheric reactive N pool, the atmospheric soluble-P pool is less studied and remains highly uncertain. Okin et al. (2011) evaluated the impact of Fe and P atmospheric deposition to the ocean in increasing $N_2$-fixation and found that Fe deposition is more important than P deposition in supporting $N_2$-fixation, while they pointed out the large uncertainty in the bioavailability of atmospherically deposited P. Benitez-Nelson's (2000) review discussed the importance of discrete pulses of P input to the oligotrophic seas that have been found to increase the phytoplankton biomass over short timescales. They also estimated that atmospheric P deposition could be underestimated by as much as 50%, when neglecting the P fraction that is soluble under acidic and high temperature conditions.

In marine ecosystems the bioavailability of P is found to depend significantly on its degree of solubility (Anderson et al., 2010). Experimentally, bioavailable P is usually considered to be the "filterable" reactive or total reactive P that passes through a 0.45 μm membrane (Maher and Woo, 1998 and references therein). Although marine organisms, such as cyanobacteria, have evolved ways to acquire P from mineral sources under P-limited conditions (Schaperdoth et al., 2007), phosphate is considered as the most bioavailable form of P (e.g. Björkman and Karl, 2003). Experiments have also shown that human-produced P-containing organics, such as organophosphorus pesticide breakdown products, can also be utilized by bacteria (Cook et al., 1978). Moreover, aerosol samples originating from combustion P sources were found to be more soluble and possibly more bioavailable than those from mineral sources (Anderson et al., 2010).

Atmospheric P has a variety of sources (Fig. 1), including mineral dust, combustion products of natural and anthropogenic origin, agricultural activities (fertilizers and insecticides), bioaerosols, volcanic emissions, sea-spray and phosphine from freshwater wetlands (Mahowald et al., 2008; Tipping et al., 2014; Wang et al., 2014; Brahney et al., 2015). The total P (hereafter TP) found in natural waters can be grouped in two major forms (Maher and Woo, 1998): 1) the particulate P (PP) and 2) the soluble P often termed dissolved P (DP). The PP mainly originates from mineral material (e.g. hydroxyapatite, brushite, fluoroapatite, variscite, stringite and wavellite) as well as P absorbed to mixed phases (e.g. clay-P, clay-organic-P and metal hydroxide-P). The DP on the other hand, includes orthophosphates (i.e. $H_2PO_4^-$, $HPO_4^{2-}$, $PO_4^{3-}$; hereafter referred

to as PO4) and inorganic condensed P (pyro-, meta- and polyphosphates). However, both PP and DP can also contain organic P (OP), of both natural and anthropogenic origin. Naturally emitted OP can be sugar-P, inositol-P, phospholipids, phosphoproteins, phosphoamides mainly associated with plants, animal and bacterial cellular materials (Maher and Woo, 1998), commonly present in atmospheric aerosols of biological origin. In addition, orthophosphate monoesters are known products of ribonucleic acid (RNA) and lipids degradation, that dominate the OP pool in the marine environment, which also contains orthophosphate diesters and phosphonates (Paytan et al., 2003).

Mineral dust has been estimated to be the largest external-to-the-ocean source of bioavailable P (Mahowald et al., 2008). These authors estimated a global P mineral source of 1.15 Tg-P yr$^{-1}$, by taking into account a typical observed P fraction of 720 ppm in dust emissions. They also applied a constant solubility fraction of 10% on the dust mineral source, based on the observations of Baker et al. (2006) for Saharan P-containing aerosols over the Atlantic ocean, in order to estimate the soluble P source associated with mineral dust. Recently published aerosol and deposition observations of African dust at Miami and Barbados (Zamora et al., 2013) suggest a total P-content of about 880 ppm, which is in the range of P fraction in dust from earlier studies (roughly 700-1090 ppm as reviewed by Mahowald et al. (2008)). Furthermore, based on OP:OC atomic ratios of 0.001-0.009 observed in several types of soils, Kanakidou et al. (2012) calculated that about 0.03 Tg-P yr$^{-1}$ of OP (10% of which is soluble) is also emitted together with soil dust in the global atmosphere.

P-containing dust solubilisation in deliquesced mineral dust aerosols is expected to significantly contribute to the soluble inorganic forms of P (DIP) in the atmosphere. Nenes et al. (2011) suggested that dissolution of apatite minerals (i.e. $Ca_5(PO_4)_3(OH,F,Cl)$) under acidic conditions can explain the observed DIP levels over the Eastern Mediterranean, a characteristic region where Saharan dust can interact with polluted air masses from Europe and the Middle East. Under acidic atmospheric conditions, $H^+$ can react with the $PO_4$ and the HO- or F- groups in the crystal surface, weakening the $Ca^{2+}$ bonds and thus phosphate to be mobilized from the crystal surface (Christoffersen and Christoffersen, 1981). Hence, mineral dust acid dissolution under polluted conditions can potentially increase the bioavailable P supply into oceanic regions and further stimulate the net primary production of marine ecosystems (Nenes et al., 2011).

Primary P sources from combustion processes of anthropogenic and biomass burning origin are estimated to contribute significantly to global P fluxes in the atmosphere (Mahowald et al., 2008; Tipping et al., 2014; Wang et al., 2014; Brahney et al., 2015). However, the estimates of global strength of the primary P combustion source vary by about an order of magnitude on the global scale, due to the consideration of different forms of the emitted P (i.e. residual or P-containing ash, gaseous or particulate P produced during combustion processes; Wang et al., 2014) and different size distributions in the emitted P-containing particulate matter. Mahowald et al. (2008) using observed mass ratios of P to black carbon (BC) for fine (<2 μm) and coarse (2 μm ≤ mean particle diameter < 10μm) particles (Mahowald et al., 2005), calculated emission fluxes from biomass burning and anthropogenic fuel (i.e. fossil fuel and biofuel) combustion of 0.03 Tg-P yr$^{-1}$ and 0.05 Tg-P yr$^{-1}$, respectively. Tipping et al. (2014) estimated a global atmospheric P emission flux of 3.7 Tg-P yr$^{-1}$ by combining observed deposition rates over land together with modelled deposition rates over the ocean. This emission flux also accounts for P deposition fluxes of larger particles (i.e. primary biological material in the aerosol mode >> 10μm) that are mainly deposited very close to their source region and thus not long-range transported. On the other hand Wang et al. (2014), by assuming that combustion processes emit significant amounts of P as large particles > 10μm (hereafter as super-coarse particles) calculated that P emissions from biomass burning and anthropogenic combustion processes can contribute about 0.7 Tg-P yr$^{-1}$ and 1.8 Tg-P yr$^{-1}$ respectively. In contrast to that study, which was more focused on the impact of anthropogenic combustion on the global P source, Brahney et al. (2015) extended the methodology of Mahowald et al. (2008) in a more explicit aerosol size manner by taking into account also the naturally emitted super-coarse P-containing

particles (i.e. dust, primary biological material and sea-salt). Brahney et al. (2015) showed that considering this super-coarse fraction as an additional P source, the estimated deposition fluxes close to the source areas where large particles are emitted (e.g. Tipping et al., 2014) can be significantly improved.

The sea-surface microlayer can also act as an atmospheric source of P in the marine environment (Graham et al., 1979). Correlations between sea-salt fluxes and seawater P concentrations revealed a 10-200 fold enrichment of P content in sea salt particles compared to sea-water Na concentrations (Graham and Duce, 1979; Graham et al., 1979). However, this enrichment was found to decrease with increasing wind velocity, introducing significant uncertainty in the strength of the oceanic flux of P on a global scale. Vet et al. (2014) by reviewing deposition observations and specifically based on inland background site measurements and trajectories analysis from the remote southern ocean, pointed out that sea spray may be a weak contributor to atmospheric P. Mahowald et al. (2008) taking into account a constant Na concentration in seawater of 10.781 g-Na kg-water$^{-1}$ and surface seawater phosphate concentrations from the NOAA Data Center, calculated a global annual flux of soluble P of 0.005 Tg-P yr$^{-1}$ (accounting for particles up to 10μm in diameter; i.e. $PM_{10}$). Wang et al. (2014), used a total oceanic emission P flux of 0.16 Tg-P yr$^{-1}$ that was calculated as the average of earlier estimates (ranging from 0.005-0.33). Additionally, Paytan et al. (2003) found that OP in the seawater particulate matter can be up to 80% of total P. Based on an OP/Na mass ratio of 0.02% as observed by Graham and Duce (1979), Kanakidou et al. (2012) estimated that the surface global ocean may also emit 0.19 – 0.80 Tg-P yr$^{-1}$ in the form of OP.

Bioaerosols are P-carriers (Mahowald et al., 2008) that can significantly contribute to the OP budget in the atmosphere (Kanakidou et al., 2012). These primary biological aerosol particles (hereafter PBAPs) usually range from 10 nm to roughly 100 μm in diameter and depending on their sizes, origin and type, can be transported over long distances. PBAPs can be either alive, dead, dormant (e.g. bacteria, viruses and fungi spores) or products released from living organisms such as pollen (Ariya et al., 2009). Mahowald et al. (2008) calculated that PBAP contribute 0.165 Tg-P yr$^{-1}$ while Kanakidou et al. (2012), based on organic carbon (OC) estimates of PBAPs emissions and by using a OP:OC atomic ratio of 0.001, calculated that PBAPs contribute about 0.13 Tg-P yr$^{-1}$ to global OP emissions. Large uncertainties, however, are associated with this estimate since it relies on the applied OP:OC ratios of PBAPs that have been observed to range over 2 orders of magnitude from about 0.0002 up to 0.02 (Kanakidou et al., 2012 and references therein) and on the simplified approximation of the density (1-1.2 g cm$^{-3}$) used for the conversion of the PBAPs number fluxes to mass units (Burrows et al., 2009a, 2009b). Mahowald et al. (2008) and Kanakidou et al. (2012) assumed half of the PBAP source to be hydrophilic, while Heald and Spracklen (2009) assumed all PBAPs to be totally hydrophilic particles using an OM:OC ratio of 2.6 that is based on observations of fungal spores as proposed by Bauer et al. (2008). However, bacteria (e.g. *P. syringae*) are considered as rather insoluble bioaerosols, in contrast to the water soluble fractions of highly polar sugars (fructose, glucose, sucrose, trehalose) and sugar alcohols (arabitol, inositol, mannitol), mainly contained in pollen grains and fungi spores (Ariya et al., 2009). Ageing during atmospheric transport is also expected to increase bioaerosols' solubility, converting a fraction of their insoluble OP content to soluble OP (DOP) due to the uptake of oxidants and the formation of larger chains of soluble multifunctional groups (Ariya et al., 2009). Regardless of bioaerosols being hydrophilic or not, because they consist of biological material, they are expected to be bioavailable (Björkman and Karl, 1994). The degree of hydrophilicity therefore is more important for determining the relative importance of dry and wet deposition during their supply to the oceans.

In the present study, the 3-D chemical transport global model TM4-ECPL is used to integrate current knowledge on the atmospheric P cycle and simulate the atmospheric concentrations and deposition fluxes of P over land and oceans, driven by mineral, natural and combustion P emissions. To our knowledge, this is the first study that accounts for both inorganic and organic forms of P and their evolution in the atmosphere. Furthermore, we also present the first global estimate of the PO4

flux due to the acid-solubilisation of dust particles. The model description and the parameterization of atmospheric acidity impact on the P-solubilisation from mineral dust aerosol in atmospheric water, together with the OP atmospheric ageing contribution to the DP global budgets were presented in Sect. 2. The calculated TP and DP global atmospheric concentrations are shown and compared to observations in Sect. 3. In Sect. 4, the importance of present day air-pollutants on

DP atmospheric deposition is investigated based on simulations using past and future anthropogenic and biomass burning emission scenarios. The contribution of bioaerosols to the bioavailable P atmospheric deposition and implications of the findings concerning the biogeochemistry of marine ecosystems are also discussed (Sect. 4). Overall, the impacts of human-driven changes on the calculated DP deposition fluxes to the global ocean are summarized in Sect. 5.

**2    Model description**

The TM4-ECPL global chemistry – transport model (Myriokefalitakis et al., 2015) simulates the oxidant chemistry accounting for non – methane volatile organics and all major aerosol components, including secondary inorganic aerosols like sulphate ($SO_4^{2-}$), nitrate ($NO_3^-$), ammonium ($NH_4^+$) calculated using ISORROPIA II thermodynamic model (Fountoukis and Nenes, 2007) and secondary organic aerosols (Tsigaridis and Kanakidou, 2007; Tsigaridis et al., 2014). The atmospheric

cycles of Fe and N in TM4-ECPL have been parameterized and evaluated in Myriokefalitakis et al. (2015) and Kanakidou et al. (2016) respectively, while uncertainties in the computed atmospheric composition associated with different emissions parameterizations have been calculated in Daskalakis et al. (2015). The model's ability to reproduce distributions of organic aerosols (Tsigaridis et al., 2014) and tropospheric ozone, ozone's precursors and aerosols have been also evaluated, against satellite and in-situ observations (Eckhardt et al., 2015; Stohl et al., 2015; Quennehen et al., 2016).

TM4-ECPL is driven by the ECMWF (European Centre for Medium – Range Weather Forecasts) Interim re–analysis project (ERA – Interim) meteorology (Dee et al., 2011). The current model configuration has a horizontal resolution of $3^o$ in longitude by $2^o$ in latitude and 34 hybrid layers in the vertical, from the surface up to 0.1 hPa, with a model time-step of 30 min. TM4-ECPL uses modal size (lognormal) distributions to describe the evolution of fine and coarse aerosols in the atmosphere. To represent phosphorus in the model, overall 32 model P-containing aerosol tracers are used of different sizes

and solubilities. In TM4-ECPL, different sources emit P-containing aerosols of different sizes represented by lognormal distributions as outlined in section 2.1. For each aerosol mode and source (Figure 1) the model accounts for total P, phosphate, insoluble and soluble OP. For the dust source it also accounts for the two P-containing minerals (fluoroapatite and hydroxyapatite) as further described in the section 2.1.1. These are individually transported, aged and deposited in the atmosphere. The 'dry' aerosol hygroscopic growth in the model is treated as a function of ambient relative humidity and the

composition of soluble aerosol components based on experimental work by Gerber (1985; 1988) and this uptake of water on aerosols changes the particle size. In addition, during atmospheric transport there are major changes in the size distribution of aerosols as a consequence of the removal of larger particles due to gravitational settling. The P-containing aerosols follow the same parameterizations, hygroscopic growth and removal processes are assumed to affect the mass median radius (i.e. size).

TM4-ECPL uses anthropogenic (including ship and aircraft emissions) and biomass burning emissions from the historical Atmospheric Chemistry and Climate Model Intercomparison Project (ACCMIP) database (Lamarque et al., 2013) for the years 1850 (hereafter PAST), 1999 and 2000 from the Representative Concentration Pathway 6.0 (RCP 6.0) emission scenario (van Vuuren et al., 2011) for the years 2001 to 2010 (year 2008 is hereafter called PRESENT) and for the year 2100 (hereafter FUTURE) that have been used for the sensitivity simulations. Details on anthropogenic and natural emissions used

for this work are provided in Myriokefalitakis et al. (2015) with the exception of mineral dust that for the present study is calculated online by the model (van Noije et al., 2014), based on the dust source parameterization of Tegen et al. (2002). The three base simulations (PAST, PRESENT and FURURE) have been performed with meteorology for the year 2008. Note however that, we have extended the present day simulation to the 11-years period from 2000 to 2010 with a spin-up time of one year (i.e. with 1999 meteorology and emissions), to cover the majority of the dates with available atmospheric observations used for model evaluation (see Sect 2.4 and Sect. 3.2).

## 2.1 Phosphorus Emissions

### 2.1.1 Phosphorus emissions from mineral dust

Apatite is the most abundant primary natural source of P in soils (Newman, 1995) compared to other low solubility P forms such as secondary metal–phosphate precipitates and organic phosphate. For the present study, apatite is assumed to be the only mineral in dust that contains P. The spatially distributed fraction of P in soils ($f_P$) from the global soil mineralogy dataset developed by Nickovic et al. (2012) is used to calculate the inorganic P-containing mineral (i.e. apatite) emissions as:

$$E_P = F_{880} \cdot f_P \cdot E_{Du} \tag{1}$$

where $E_{Du}$ is the on-line calculated dust emissions in the model, $F_{880}$ is a factor applied to adjust the P emissions to the global mean P content of mineral dust in the model domain of 880 ppm per weight as observed by Zamora et al. (2013), and $E_P$ is the resulted inorganic P emissions from mineral dust. P-containing minerals associated with dust particles are emitted in the fine and coarse mode with mass median radii (lognormal standard deviation) of 0.34 μm (1.59) and 1.75 μm (2.00), respectively. The P-containing dust aerosol emissions treated as a log-normal distribution with a dry mass median radius and sigma same as that of dust particles and changes of the particle size based on the hygroscopic growth as a function of ambient relative humidity and the composition of soluble aerosol components (Gerber, 1985). Note, however, that no coagulation among different dust modes is considered for the current study.

Although in most relevant modelling studies airborne P-containing dust particle emissions are assumed to have an average P content of 720 ppm (Mahowald et al., 2008; Wang et al., 2014; Brahney et al., 2015), in the atmosphere due to transport, ageing and deposition processes the overall mineralogy may change the chemical composition and size of dust aerosol population. In a recent iron modelling study however (Perlwitz et al., 2015), a significant effort has been made to model the mineral composition of dust considering the differences from the original soil composition. Perlwitz et al. (2015) have found significant overestimate (a factor of 10-30) mainly in the fine aerosol emissions that are the smallest part of dust emissions (e.g. about 7% of the total emissions in our model) and an underestimate in the larger particles emissions both for total dust and for individual minerals when the mineralogy of dust aerosol is assumed to be the same as that of the soil. However for the present study, we did not account for different P content for dust particles in the fine and the coarse mode, since the global soil mineralogy dataset used (Nickovic et al., 2012) does not provide any information of P content in silt and clay soil particles separately. Note also, that recent studies indicate that dust super-coarse particles can be very important for the biogeochemistry over land, since they can represent the dominant fraction of dust close to source regions (Lawrence and Neff, 2009; Neff et al., 2013). Brahney et al. (2015) modelling study that focused on the atmospheric phosphorus deposition over global alpine lakes, based on Neff et al. (2013) observations, estimated that only 10% of the mass that travels in the atmosphere is within the <10 μm size fraction. In our study we do not account for super-coarse dust particles because due to their short atmospheric lifetime, they are emitted and deposited in the same model grid box (Brahney et al., 2015). This omission is not expected to have significant impact on our results, since the present work is focused on the P-solubilisation

mechanisms occurring via atmospheric long-transport mixing and on the bioavailable P deposition over the marine environment.

For the year 2008 the mineral dust emissions calculated in TM4-ECPL amount to 1181 Tg yr$^{-1}$ and the corresponding apatite emissions to 1.034 Tg-P yr$^{-1}$ with 10% of it (0.103 Tg-P yr$^{-1}$) in the soluble form (Table 1). The soluble fraction used in our model is based on the measurements of leachable inorganic phosphorus (LIP) for Saharan soil dust, as presented by Nenes et al. (2011). These authors found that LIP represented up to 10 % of total inorganic P in Saharan soil samples and dry fallout collected during Sahara dust storms before acid-treatment. Moreover, Yang et al. (2013) estimated the labile inorganic P in the top soil on the global scale at about 3.6 Pg-P that corresponds to about 10% of the estimates of total soil P on the global scale 30.6-40.6 Pg-P (Smil, 2000; Wang et al., 2010; Yang et al., 2013). To further investigate uncertainties associated with the soluble fraction of P-containing dust aerosol emissions in our model, an additional simulation has been performed neglecting any soluble fraction on initial emissions.

In addition to the desert dust inorganic P source, we account for the OP present in soil's organic matter, following the method developed by Kanakidou et al. (2012 and references therein). Thus, using a mean OP:OC molar ratio of 0.005, a mean OM content of soil dust of 0.25% and an OM:OC molar ratio of 1.76, we here evaluate the dust source of OP at 0.022 Tg-P yr$^{-1}$ for the year 2008. This flux is in good agreement with the 0.03 Tg-P yr$^{-1}$ calculated for 2005 by Kanakidou et al. (2012) using the same methodology but with the AEROCOM database for dust emission fluxes (Dentener et al., 2006). Note that similarly to that earlier study, a solubility of 10% is here applied to the OP dust emissions.

### 2.1.2 Phosphorus emissions from combustion sources

For the present study, the P/BC mass ratios of combustion sources as estimated by Mahowald et al. (2008) (i.e. 0.0029 for fine aerosols and 0.02 for coarse aerosols) are applied to the inventories of monthly BC emissions of anthropogenic (i.e for fossil fuel, coal, waste and biofuel) and biomass burning origin, as provided by the historical ACCMIP database for 1850 and from the RCP6.0 for 2008 and 2100. In the model, a number mode radius of 0.04 μm and a lognormal standard deviation of 1.8 are assumed for fine P emissions, while for coarse P a number mode radius of 0.5 μm and lognormal standard deviation of 2.00 are used as proposed for combustion aerosols by Dentener et al. (2006). BC emissions from anthropogenic combustion in the coarse mode are assumed to be 25% of those in the fine mode (Jacobson and Streets, 2009), while biomass burning emissions in the coarse mode are assumed equal to 20% of those of fine aerosols (Mahowald et al., 2008). Thus, the computed anthropogenic combustion and biomass burning annual mean sources of TP are calculated to be 0.043 Tg-P yr$^{-1}$ (by about 70% in the coarse mode) and 0.018 Tg-P yr$^{-1}$ (by about 66% in the coarse mode) respectively, all corresponding to the year 2008. Despite the different emission databases and the aerosol size parameterization, the computed present-day TP sources for the year 2008 are comparable to those of Mahowald et al. (2008) for the year 2000 (i.e. 0.045 Tg-P yr$^{-1}$ and 0.025 Tg-P yr$^{-1}$ for anthropogenic combustion and biomass burning, respectively). PAST, PRESENT and FUTURE combustion emissions calculated for this study based on the ACCMIP and RCP 6.0 database are presented in Table 1.

Half of TP emissions from combustion sources are considered to be in the form of OP following the approach of Kanakidou et al. (2012). All P-containing particles from combustion emissions are here treated initially as 50% soluble (Mahowald et al., 2008). The insoluble fraction of OP associated with combustion emissions can be further converted to soluble OP (DOP) during atmospheric ageing, using the ageing parameterization for primary hydrophobic organic aerosols in the model (Tsigaridis and Kanakidou, 2003; Tsigaridis et al., 2006), but for the respective size and lognormal distribution of OP aerosols with the larger particles experiencing the smallest conversion rates.

To further investigate uncertainties in the P combustion emissions in our model, an additional present-day simulation was performed taking into account the total (bulk) mass of anthropogenic combustion and biomass burning P emissions, as developed by Wang et al. (2014) (R. Wang, personal communication, 2016). According to that database, global anthropogenic emissions from fossil fuels, biofuels and deforestation fires amount to 1.079 Tg-P yr$^{-1}$ and natural fire emissions equal to 0.808 Tg-P yr$^{-1}$. For this sensitivity simulation, we apply the size distribution as described in Wang et al. (2014); i.e. by dividing total emissions into 3 modes - one fine (2% of P) and two coarse modes (25% and 73% of P) - with mass mode dry diameters of 0.14 μm, 2.5 μm and 10 μm and lognormal standard deviations of 1.59 and 2.00 for fine and coarse modes, respectively.

### 2.1.3    Phosphorus emissions from primary biological aerosol particles

Three types of P-containing PBAPs are considered for the present study: bacteria (BCT), fungal spores (FNG) and pollen grains (PLN). PBAPs from other sources, such as insect fragments and plant debris (e.g. Després et al., 2012), are however neglected in the present study. Omission of these super coarse particles is expected to lead to an underestimate in the PBAPs contribution to P deposition over land that requires to be evaluated with targeted observations. The BCT fluxes are parameterized based on the Burrows et al. (2009) best-fit estimates for particles of 1 μm diameter flux rates (*f*) and for six different ecosystems: coastal: 900 m$^{-2}$ s$^{-1}$, crops: 704 m$^{-2}$ s$^{-1}$, grassland: 648 m$^{-2}$ s$^{-1}$, land-ice: 7.7 m$^{-2}$ s$^{-1}$, shrubs: 502 m$^{-2}$ s$^{-1}$ and wetlands: 196 m$^{-2}$ s$^{-1}$. For the present study, the Olson Global Ecosystem Database (Olson, 1992), originally available for 74 different land types on a spatial scale of 0.5° x 0.5°, is lumped into 10 ecosystem groups as proposed by Burrows et al. (2009). The total BCT flux ($F_{BCT}$; s$^{-1}$) in the model is calculated based on the aforementioned fluxes ($f_i$; m$^{-2}$ s$^{-1}$) per ecosystem (*i*), weighted by the respective ecosystem area fraction in the model gridbox ($a_i$; m$^2$), as:

$$F_{BTC} = \sum_{i=1}^{6} a_i \cdot f_i \tag{2}$$

Heald and Spracklen (2009) proposed that FNG fluxes linearly depend on the leaf area index (LAI; m$^2$ m$^{-2}$) and the specific humidity (q; kg kg$^{-1}$), based on near-surface mannitol observations. For the present study however, we use a recently published emission parameterization proposed by Hummel et al. (2015), as derived based on fluorescent biological aerosol particles field measurements at various locations across Europe and for spores with a mean dry diameter of 3μm (eq. 3):

$$F_{FNG} = 20.426 \cdot (T - 275.82\,K) + 3.93 \cdot 10^4 \cdot q \cdot LAI \tag{3}$$

In the TM4-ECPL that parameterization (eq. 3) is used to calculate FNG emissions online, using monthly averaged LAI distributions and 3-hourly averaged specific humidity (q) and temperature (T) data, as provided by the ERA-Interim.

PLN emissions maximize when plant surfaces are dry, under high turbulence during the morning hours and during spring months (Jacobson and Streets, 2009). Hoose et al. (2010) parameterised the pollen flux rate as linearly dependent on LAI assuming particles with a mean dry diameter of 30μm, by simplifying the more sophisticated parameterisation developed by Jacobson and Streets (2009) for a global model. Here, we use the Jacobson and Streets (2009) pollen parameterization (particle mean dry diameter of 30 μm), with the number pollen flux ($F_{PLN}$; s$^{-1}$) calculated by the following equation:

$$F_{PLN} = f_{PLN} \cdot LAI \cdot R_{month} \cdot R_{hour} \tag{4}$$

where, $f_{PLN} = 0.5$ m$^{-2}$s$^{-1}$, the factor $R_{month}$ accounts for the seasonal and $R_{hour}$ the hourly pollen flux variation.

PBAPs are here assumed to be monodisperse spherical particles (Hoose et al., 2010; Hummel et al., 2015) of 1 g cm$^{-3}$ density (Sesartic and Dallafior, 2011) with an organic matter to organic carbon (OM:OC) ratio set equal to 2.6 (i.e. that of mannitol)

corresponding to a molecular weight equal to 31 g mol⁻¹, as suggested by Heald and Spracklen (2009). According to our model estimates roughly 60 Tg-C yr⁻¹ are emitted as PBAP. Bacterial emissions are assumed as completely insoluble (Ariya et al., 2009), fungal spores are emitted as 50% soluble aerosols (Mahowald et al., 2008; Kanakidou et al., 2012), while pollen are emitted as totally soluble aerosols (Hoose et al., 2010). A constant mean P:C atomic ratio of 0.001 is used for PBAPs, as suggested by Kanakidou et al. (2012) and all P is assumed in the form of OP. Based on the above parameterizations the model calculates an OP emission flux associated with PBAP equal to 0.156 Tg-P yr⁻¹, of which 0.123 Tg-P yr⁻¹ (about 80%) are considered to be in the form of DOP (Table 1). However, because PBAPs consist of biological material they are here considered to be bioavailable for marine ecosystems, as further discussed in Sect. 4.1 and Sect.4.2. In addition in TM4-ECPL, upon emission the insoluble fraction of PBAPs becomes progressively soluble due to atmospheric ageing. This process that has been seen to occur for instance by degradation of RNA (Paytan et al., 2003), in TM4-ECPL is parameterised based on oxidant levels as for all organic aerosols (Tsigaridis and Kanakidou, 2003; Tsigaridis et al., 2006).

### 2.1.4    Phosphorus emissions from sea-spay

Oceanic P emissions associated with sea-spray are here computed on-line based on a sea-salt emission flux parameterization of Vignati et al. (2010), accounting for fine and coarse modes, with number mode dry radii of 0.09 μm and 0.794 μm, and lognormal standard deviations of 1.59 and 2.00 for accumulation and coarse particles, respectively. Sea-spray emissions are driven by the model's meteorology and for the year 2008 the model calculates a total of about 8284 Tg yr⁻¹ of sea-salt emissions (of which 41 Tg yr⁻¹ are in the fine mode). These numbers compare well with the AEROCOM recommendation of 7925 Tg yr⁻¹ by Dentener et al. (2006) and are within the range of 2272-12462 Tg yr⁻¹ computed by Tsigaridis et al. (2013) using several different parameterisations. Note that our sea-salt source estimation is however much lower than the one used in the modelling study by Wang et al. (2014) (i.e. 25300 Tg yr⁻¹), since super coarse sea-salt particles are not considered in the current parameterization.

The oceanic P emissions in TM4-ECPL are calculated as:

$$E_{PO_4} = \frac{[P]/MW_P}{[Na]/MW_{Na}} \cdot E_{Na} \qquad (5)$$

where [P] is the P seawater concentrations in μM, [Na] is Na seawater concentration in μM and $E_{Na}$ is the sea-salt emission flux from the ocean surface in kg-Na m⁻² s⁻¹. MW is the corresponding molecular weight of P and Na, used to convert molar to mass ratios. In TM4-ECPL, sea-salt particles are emitted from the ocean's surface every time-step using surface wind-speed data from the ERA-Interim database (updated every 3 hours). Surface seawater $PO_4$ concentrations come from the LEVITUS94 World Ocean Atlas (Conkright et al., 1994; http://iridl.ldeo.columbia.edu/SOURCES/.LEVITUS94/.ANNUAL/.PO4/) ranging up to about 3 μM of $PO_4$ in the global ocean. Taking into account that the average Na concentration in seawater is about 10.781 g-Na kg-water⁻¹ and an average seawater salinity of 35.5 g kg-water⁻¹, the spatial distribution of surface oceanic Na concentrations can be derived from the distribution of the surface salinity concentrations as provided by the LEVITUS94 World Ocean Atlas (Levitus et al., 1994; http://iridl.ldeo.columbia.edu/SOURCES/.LEVITUS94/.ANNUAL/.sal/). Note that surface concentrations, both for seawater $PO_4$ and salinity, correspond to the data available for 0m depth (with the next available depth in the LEVITUS94 database to be at 10m).

We additionally take into account the OP oceanic emissions, as described in Kanakidou et al. (2012; see supplementary material and refferences therein). For this, the model accounts for a mean seawater OP concentration of 0.2 μM of P, based on Björkman and Karl (2003) observations. Since, to our knowledge, no spatial distribution of seawater OP concentrations is

available, the monthly mean surface chlorophyll a (Chl a) concentrations from MODIS retrievals, used in the model to derive marine primary organic aerosol emissions (Myriokefalitakis et al., 2010), are used as a proxy to geographically distribute the mean seawater OP concentrations. Overall, the model calculates an emission flux of TP equal to 0.008 Tg-P yr$^{-1}$ from the global ocean (Table 1), of which 0.001 Tg-P yr$^{-1}$ is in the form of OP. Note that the insoluble fraction of oceanic OP in the model can be transferred to the soluble mode (DOP) due to atmospheric ageing processes. The omission of the super coarse sea salt aerosol might affect our estimates of P deposition to the ocean. Brahney et al (2015) evaluated this source at 0.0046 Tg-P yr$^{-1}$, an amount that introduces a 3% underestimate to the here calculated present-day P deposition flux to the oceans.

### 2.1.5     Phosphorus emissions from volcanic aerosols

Mahowald et al. (2008) estimated that about 0.006 Tg-P yr$^{-1}$ are associated with volcanic aerosols on a global scale, based on volcanic plume observations. Although on a global scale, volcanic ash is a small source of TP, it is found to impact, at least regionally, the ocean nutrients distributions and marine productivity (Uematsu et al., 2004; Henson et al., 2013; Olgun et al., 2013). For the present study, we applied that global annual mean volcanic flux (see also Table 1), using the distribution of sulphur volcanic emissions by Andres and Kasgnoc (1998) as updated by Dentener et al. (2006). Volcanic phosphorus is here assumed to reside in the fine particulate mode and is treated in the model as totally soluble aerosol (i.e. DIP), as proposed by Mahowald et al. (2008). The log-normal size-distribution parameters used for volcanic P aerosol are a number mode radius of 0.04 μm and a lognormal standard deviation of 1.8 (Dentener et al. (2006) for sulphate fine aerosols from continuous volcanic eruptions).

### 2.2     Phosphorus acid-solubilisation mechanism

Phosphorus solubilisation from mineral dust under acidic atmospheric conditions, is here assumed to occur for the least- and the most-soluble member of apatite minerals as proposed by Nenes et al. (2011): the fluorapatite ($Ca_5(PO_4)_3(F)$; hereafter FAP) and the hydroxyapatite ($Ca_5(PO_4)_3(OH)$; hereafter HAP), respectively. FAP is considered as a geologically abundant apatite, usually present in the form of igneous or sedimentary carbonate FAP (Guidry and Mackenzie, 2003). However, due to lack of information on the relative abundance and geographic distribution of FAP and HAP in soils, we here assume equal mass fractions of FAP and HAP in apatite containing soils.

The dissolution of FAP and HAP here is treated as a kinetic process, the rate of which depends on the H$^+$ activity of atmospheric water (i.e. aerosol water and cloud droplets), the reactivity of P species, the ambient temperature and the degree of solution saturation. For aerosol water, the activity of H$^+$ is calculated on-line in the model by the thermodynamic module ISORROPIA II (Fountoukis and Nenes, 2007). For cloud water, the H$^+$ concentration is calculated by the aqueous-phase chemistry module as presented in Myriokefalitakis et al. (2011; 2015). The phosphate dissolution rate ($R$), as moles of $HPO_4^{-2}$ per second per gram of apatite, is obtained using the empirical formulation of Lasaga et al. (1994):

$$R = K(T) \cdot a(H^+)^m \cdot f \cdot A \tag{6}$$

where $K$ is the reaction constant in moles m$^{-2}$ s$^{-1}$, $a(H^+)$ is the H$^+$ activity, $m$ is the experimentally derived reaction order with respect to the solution H$^+$ concentration, and $A$ is the specific surface area of each apatite-containing particle in m$^2$ g$^{-1}$. The function $f$ (Cama et al., 1999) depends on the solution saturation state ($0 \le f \le 1$) and is given by:

$$f = 1 - Q / K_{Eq} \tag{7}$$

where, $Q$ is the reaction activity quotient, $K_{Eq}$ is each apatite equilibrium constant and $Q/K_{eq}$ is the fraction that expresses the state of saturation of the solution (with respect to the apatite), calculated every timestep in the model. Thus, when $f = 1$, the solution is far from equilibrium, therefore the dissolution rate becomes maximum; while as $f$ approaches 0, the solution approaches equilibrium with any remaining undissolved FAP and HAP.

HAP is experimentally found to be roughly 3 orders of magnitude more soluble than FAP ($K_{Eq}$(HAP) = $10^{-20.47}$ vs. $K_{Eq}$(FAP) = $10^{-23.12}$), as reported by Nenes et al. (2011) based on van Cappellen and Berner (1991). According to the compilation of experimental determinations of P-dissolution rates of HAP and FAP by Palandri and Kharaka (2004), the dissolution rate of HAP is found to be about an order of magnitude slower than that of FAP under highly acidic conditions (K(HAP) = $10^{-4.29}$ and K(FAP) = $10^{-3.73}$ for pH=0), while under neutral conditions, HAP is found to dissolve two orders of magnitude faster

than FAP (K(HAP) = $10^{-6}$ and K(FAP) = $10^{-8}$ for pH=7). Moreover, HAP is measured to have almost 8 times larger specific surface area (80.5 $m^2$ $g^{-1}$, Bengtsson et al., 2009) compared to that of FAP (10.7 $m^2$ $g^{-1}$), which is in agreement with the measured specific surface areas of 8.1-16 $m^2$ $g^{-1}$ for sedimentary FAP (Guidry and Mackenzie, 2003). Guidry and Mackenzie (2003) have experimentally derived different rate constants (K) for FAP dissolution ranging from 5.75 $10^{-6}$ mol $m^{-2}$ $s^{-1}$ to 6.53 $10^{-11}$ mol $m^{-2}$ $s^{-1}$ with a pH ranging from 2 to 8.5. They further derived the respective reaction orders ($m$) for each pH-

region, between 0.01 (for neutral to basic conditions) and 0.81 (for acidic conditions), while the activation energy of the FAP dissolution ($E_a$) was calculated equal to 8.3 kcal $mol^{-1}$. For the present study, the dissolution reaction coefficient K for FAP (Table 2), is based on the dissolution experiments by Guidry and Mackenzie (2003), for a range of pH (2-12), temperatures (25–55°C) as well as for various solution saturation states and ionic strengths.

    Bengtsson et al. (2009) have experimentally studied the solubility and the surface complexation of non-stoichiometric

synthetic HAP, identifying three distinct pH-regions for their batch dissolution experiments: 1) under acidic pH (<4.5) HAP dissolution is relatively high, producing high concentrations of $Ca^{2+}$ and $H_2PO_4^-$; 2) under basic pH (>8.2) surface complexation is the main process and 3) for intermediate pH (4.5-8.2) where both dissolution and surface complexation occur. However, they do not provide sufficient information to enable parameterising HAP dissolution similarly to FAP dissolution. Therefore, for HAP dissolution kinetics we use the dissolution rates of FAP after correcting them to account for

the differences between HAP and FAP dissolution kinetics as a function of pH and T, as reported by Palandri and Kharaka (2004). For this, we consider the different dissolution rates for a pH range of 0 to 7-8, which is the range of acidity encountered by atmospheric particles, including dust (e.g. Bougiatioti et al., 2016; Weber et al., 2016). At the strongly acidic limit (25°C and pH = 0), the dissolution rate of HAP is here assumed to be about 27% (i.e. $10^{-0.56}$ times) slower than that of FAP, but for neutral and basic conditions (and 25°C) HAP dissolves two orders of magnitude faster than FAP (Palandri and

Kharaka, 2004). The dissolution rate also changes with temperature; we assume that HAP dissolution has a similar activation energy to FAP (Palandri and Kharaka, 2004; Guidry and Mackenzie, 2003). Additional details for the FAP and HAP mineral dissolution rate parameters are presented in Table 2.

## 2.3     Observation data for model evaluation

    The evaluation of the global atmospheric P cycle for the present study has been performed based on available observations

of aerosol concentrations (Table S1) and deposition fluxes (Table S2) from various locations around the globe (cruises and land-based stations). The methodological details of the observations used for this study are well documented in the literature and thus are not reviewed here in detail. For DP concentrations in ambient aerosols, we compiled cruise observations of PO4 over the Atlantic Ocean (50°N–50°S) from Baker et al. (2010), over the Western Pacific (25°N–20°S) from Martino et al. (2014) and over the Eastern Tropical North Atlantic Ocean (58°S–35°N, 14°–38°W) from Powell et al. (2015). For these

oceanic cruise observations, samples were either collected separating into fine- (aerodynamic particle diameter < 1μm) and coarse-mode (1μm< aerodynamic particle diameter) particles using cascade impactors that may include or exclude particles with diameters larger than 10 μm, or using a single bulk filter. We additionally use average PO4 concentrations (aerodynamic particle diameter < 10μm) from cruise measurements over Bay of Bengal and the Arabian Sea (Srinivas and Sarin, 2012). Finally, we also took into account land-based TP and PO4 aerosol concentrations measurements from two sites in the Mediterranean i) from the Finokalia monitoring station (35$^o$20`N, 25$^o$40`E) located in the Eastern Mediterranean (Crete, Greece) and ii) from Ostriconi (42$^o$40`N, 09$^o$04`E) located in the Western Mediterranean (Corsica, France). The samples at both sites were collected either separating for the fine- (aerodynamic particle diameter < 1.3 μm) and the coarse-mode (10 μm > aerodynamic particle diameter > 1.3 um) (Koulouri et al., 2008; Mihalopoulos and co-workers, unpublished data) or as bulk (Markaki et al. 2010). Details about the characteristics of these Mediterranean sampling sites can be found in Markaki et al. (2010), while the methodology for aerosol sampling and analysis is described in detail in Koulouri et al. (2008).

Although P deposition fluxes data are rather limited on a global scale, for the present study we use the wet and dry deposition fluxes (both for TP and DP) compiled by Vet et al. (2014) (R. Vet, personal communication, 2016). For wet deposition of DP, we use available filtered (i.e. analyzed as orthophosphates with no digestion as DIP) and unfiltered (i.e. analyzed as orthophosphates following digestion as total DP) annual measurements (Fig. 8.2 in Vet et al., 2014). For the TP wet deposition measurements we use annual wet deposition measurements (Fig. 8.3 in Vet et al., 2014) of unfiltered samples. The compilation of the phosphorus dry deposition fluxes by Vet et al. (2014) is based on airborne phosphorus (TP and PO4) concentrations from around the world and gridded annual dry deposition velocities from the Mahowald et al. (2008) modelling study (Fig. 8.6 and Fig. 8.7 in Vet et al., 2014). The size distribution used in these dry deposition calculations, is the same as in the modelling study by Mahowald et al. (2008), thus the derived dry deposition fluxes account for particles with diameter up to 10 μm. Finally, we also take into account DP wet and dry deposition observations from the Finokalia Station in the Eastern Mediterranean (Markaki et al., 2010; Mihalopoulos and co-workers, unpublished data), based on rain water samplings (wet only collector) and glass-bead devices respectively. Further details on the methodology of the deposition measurements at Finokalia can be found in Markaki et al. (2010).

# 3    Results and Discussion

## 3.1    Sources of atmospheric phosphorus

Figure 2 presents the annual mean primary TP and DP emissions from the various sources taken into account in the model (in the supplement the emission distribution per source for TP and DP are also presented in Fig. S1 and Fig. S2, respectively). TP emissions (Fig. 2a) maximize over the major deserts of the world (e.g. Sahara, Gobi, Arabian, Kalahari, North American and Australian deserts) with simulated P fluxes up to 100 ng-P m$^{-2}$ s$^{-1}$ (Fig. 2a and Fig. S1a). Secondary maxima of TP emission fluxes of about 0.1-1 ng-P m$^{-2}$ s$^{-1}$ are also calculated over the mid-latitudes of the northern hemisphere (NH), such as China, Europe and the US, due to release to the atmosphere of TP in ash produced during combustion processes of anthropogenic origin (Fig S1b) and over forested areas in equatorial America. Additionally, during biomass burning episodes TP is further released to the atmosphere (Fig S1c), however at rates about one order of magnitude lower than those of combustion of anthropogenic origin (roughly 0.01 ng-P m$^{-2}$ s$^{-1}$).

The same pattern (as for TP emissions) is simulated for the P soluble fraction (Fig. 2b), but with lower emission fluxes (e.g. about 1 ng-P m$^{-2}$ s$^{-1}$ over the Sahara Desert). This is attributed to the solubility of P-containing mineral dust at emission that corresponds to the DP present in the desert soil due to weathering. As discussed in Sect. 2.1.2, this fraction is taken equal to 10% for the present study. Associated mineral DP emissions (Fig. S2a) of 0.106 Tg-P yr$^{-1}$ (as PO4 and/or DOP) occur

mainly over the Saharan desert region, but significant fluxes are also calculated to occur over other important deserts of the globe. Anthropogenic DP emissions (0.021 Tg P yr$^{-1}$) occur mainly over densely populated regions of the globe (e.g. the mid-latitudes of the NH; such as China, Europe and the US), with simulated fluxes up to 0.1 ng-P m$^{-2}$ s$^{-1}$ (Fig. S2b). DP emissions from biomass burning contribute about 0.009 Tg-P yr$^{-1}$, peaking over intense biomass burning areas, e.g. tropical and high latitude forests and showing maxima over Central Africa, Indonesia and Amazonia (Fig. S2c).

The present day annual apatite dissolution flux is calculated equal to 0.444 Tg-P yr$^{-1}$ (Table 3, Fig 2c). Most of the apatite dissolution fluxes occur downwind of the major dust source regions (i.e. Nigeria downwind of the Sahara Desert, Pakistan downwind of the Thar Desert and China downwind of the Gobi desert). Over these regions, the long- and regional- range transport of natural and anthropogenic pollutants enhance atmospheric acidity and subsequently P is mobilized from mineral apatite. The model calculates maximum dissolution fluxes downwind of the Sahara and Gobi Deserts, over the Persian Gulf,

the whole Middle East and the Mediterranean basin as well as over the equatorial Atlantic. In addition, enhanced apatite dissolution is calculated over the tropical Atlantic Ocean, India and the outflow of Asia to the Pacific Ocean, in line with observations of changes in solubility during transport of dust across the tropical Atlantic Ocean by Baker et al. (2006a).

As explained in Sect. 2, for the present study the apatite dissolution (Fig. 2c) is due to the respective FAP and HAP solubilisations that occur both in aerosol water and cloud droplets (Fig. S3). The model calculates that most of the apatite

dissolution (0.111 Tg-P yr$^{-1}$) is occurring in deliquesced particles (Fig. 3; S3a and b), mainly attributed to the higher aerosol acidity, while only 0.034 Tg-P yr$^{-1}$ are calculated to occur in cloud droplets (Fig. 3b; Fig. S3c and d). Note that the model-calculated global mean pH in clouds is about 4.5 (Myriokefalitakis et al., 2015). In addition, the distributions of aerosol and cloud dissolution of apatite are rather different (Fig. 3a,b). In-cloud dissolution is calculated to maximize i) off-shore the African continent (i.e. over Cote d'Ivoire, Nigeria and Cameroon) over the equatorial Atlantic Ocean and ii) over China and

India, where dust aerosols downwind of major desert regions (i.e. Sahara and Gobi Desert respectively) meet polluted and acidic cloud droplets; while dissolution in aerosol water shows also high rates over US, Europe and Saudi Arabia.

DIP fluxes from HAP dissolution in the cloud droplets (Fig. S3d) are calculated to be roughly 60% higher than those of FAP (Fig. S3c; 0.021 Tg-P yr$^{-1}$ against 0.013 Tg-P yr$^{-1}$). However for the DIP dissolution fluxes from the FAP and the HAP in aerosol water, no differences are calculated (Fig S3a and b) for the more acidic environmental in the fine aerosol water

(0.015 Tg-P yr$^{-1}$ for each of them). On the contrary, the HAP is more soluble than FAP in the less acidic coarse aerosol water (0.041 Tg-P yr$^{-1}$ for the HAP compared to 0.039 Tg-P yr$^{-1}$ for the FAP) (see also Fig. S2 in Myriokefalitakis et al. (2015) supplementary material for pH calculations in the model). The changes in the saturation factor (*f*) in the aerosol water are also of importance. Under conditions with HAP more soluble than FAP, the respective mobilized PO4 concentrations increase faster in the aerosol solution and react with the soluble Ca$^{2+}$ present in dust, ultimately forming amorphous apatite

that precipitates from the solution (i.e. *f*=1, thus the dissolution process stops). In the presence of soluble Ca$^{2+}$ and PO4, other salts, such as monenite (CaHPO$_4$) (Somasundaran et al., 1985), can also be formed and further impact the solution's degree of saturation. These results suggest that the solution saturation effect in dust aerosol water can be a critical control on the observed PO4 enhancement in acidic atmosphere conditions.

Finally, a significant amount of DOP (0.032 Tg-P yr$^{-1}$) is added to the total DP sources due to the ageing of OP-containing

aerosols during atmospheric transport (Table 3). This amount corresponds to about 12% of the global DP primary emission

sources and to roughly 22% of the total dust-P acid solubilisation flux on a global scale. The ageing of OA carrying P presents maxima over forested areas (about 0.1 ng-P $m^{-2}$ $s^{-1}$) due to the high oxidation of PBAP (Fig. 2d). Secondary maxima are also calculated over China (0.01-0.1 ng-P $m^{-2}$ $s^{-1}$) attributed to ageing of primary OP of anthropogenic origin. Downwind of desert source regions significant DOP production rates, up to 0.1 ng-P $m^{-2}$ $s^{-1}$, are calculated over the Sahara, the Thar and
Gobi Deserts; however these DP formation rates are more localized over continental regions than those due to acid solubilisation mechanism of the dust mineral content (Fig. 2c). Non-negligible, however, DOP production is also calculated over the coastal oceans, owing to the OP ageing under the long-range transport in the atmosphere.

## 3.2      Evaluation of phosphorus simulations

Figure 4 presents the evaluation of present day model simulation at various locations around the globe (Fig a; see also Sect.
2.4), against 1) P-containing aerosol airborne concentrations (Fig. 4c,e) and 2) dry deposition fluxes (Fig. 4d,f). PO4 and TP aerosol concentrations are provided in a daily resolution (except for TP concentrations from the Corsica Island which are provided as monthly means) and for different sizes; i.e. fine ($PM_1$ or $PM_{1.3}$) and coarse ($PM_1$-to-$PM_{10}$) or $PM_{10}$ aerosols or as bulk concentrations (Table S1). For this model evaluation, a point-by-point comparison has been performed accounting for the respective daily (or monthly) outputs and aerosol size of each P-containing aerosol component of our model to the
corresponding observation database. The normalized mean bias (NMB) for the statistical analysis is calculated as:

$$NMB = \frac{\sum_{i=1}^{N}(M_i - O_i)}{\sum_{i=1}^{N} O_i} \times 100 \text{ \%} \tag{8}$$

where, $O_i$ and $M_i$ stand for observations and model predictions respectively, with N to represent the number of pairs (observations, model predictions) that are compared. More information about the model performance per database (cruise and stations) and aerosol size can be found in Fig. S4.

The comparison of all available DP aerosol measurements (fine, coarse and bulk) with the respective model results is presented in Fig. 4c. DP aerosol concentrations from cruise observations are in the range of about 3.1 $10^{-6}$ – 4.03 $10^{-2}$ μg-P $m^{-3}$ while from stations observations this range is about 3.23 $10^{-4}$ – 1.37 $10^{-1}$ μg-P $m^{-3}$. The model overestimates the DP cruise observations (NMB = 21%) and underestimates the DP concentrations measured at stations (NMB = -84%). Focusing however on the size- segregated comparison of aerosol DP (Fig. S4), the model underpredicts the observed concentrations at
the Finokalia station, both for fine and coarse particles, implying thus a respective underestimation of P sources over land. On the contrary, for cruise measurements the model performs much better both for fine and coarse aerosols as well as bulk observations. Note that the station observations correspond to those of Finokalia and Corsica. Furthermore, only few cruise TP observations are available (Graham and Duce, 1982; Baker et al., 2006a; Baker et al., 2006b) that are discussed later in section 3.5. The here presented comparison also indicates that the model underpredicts (NMB = -59%) the observed TP
concentrations at Finokalia (Eastern Mediterranean) (see also Fig. S4), however it simulates better the bulk TP aerosol concentrations at Corsica (Western Mediterranean). This implies that our model lacks TP sources in the Eastern Mediterranean atmosphere, which is strongly affected by air masses from surrounding regions and by sources other than local ones.

As in the case of DP aerosol concentrations, the model simulates better (NMB = 52%) the DP dry deposition fluxes over
oceanic regions (airborne cruise measurements compiled by Vet et al. (2014)) than the observations (NMB = -93%) at the Finokalia station (Fig 4d). Note that the same pattern is also calculated for the TP dry deposition fluxes (Fig. 4f). The

omission of super-coarse marine DP sources associated with sea-salt particles can explain some discrepancies between model results and observations only when these later concern bulk aerosols in oceanic regions (so they could include super-coarse particles), which is the case for wet or dry deposition samples. As discussed in Sect. 2.1.4, this omission can affect local comparisons but overall does not introduce more than a 3% underestimate of DP flux over the ocean. In many cases, aerosol samples have been collected with inlet devices that enable collection of specific fractions of aerosols and eliminate super-coarse particles. When bulk aerosols have been collected, then the presence of super-coarse aerosols might introduce discrepancies between model results and observations. Overall the model performs better for DP dry deposition fluxes over the oceans than over land, indicating a possible underestimate in the continental source of P.

In Figures S4 and S5 (supplement) are also presented the results of sensitivity simulations and the base case simulation with the aerosol observations and dry and wet deposition fluxes, respectively. Fig. S6 also shows the comparison of the annual cycles of the atmospheric concentrations (TP and PO4) and deposition fluxes (dry and wet deposition), against the TM4-ECPL monthly model results. For cruise measurements over the Atlantic and Pacific Oceans (Baker et al., 2010; Martino et al., 2014; Powell et al., 2015) and the global compilation of deposition rates (Vet et al., 2014), the observations are also spatially averaged inside the same model grid box. These comparisons show almost similar performance for all sensitivity simulations but one falling in most cases close to the lower edge of observed concentrations and deposition fluxes. However, taking into account the Wang et al. (2014) P-combustion sources, the model performs better over the land (e.g. for TP concentrations at Corsica; Fig. S4g, and for DP concentrations at the Finokalia monitoring station; Fig. 6b,f,i), indicating that the base simulation underestimates either anthropogenic combustion sources or other natural P sources. Neglecting the P dissolution definitely degrades the comparisons of model results with observations. On the other hand the results show very small sensitivity to the assumption of soluble fraction of the primary emissions of P. This finding supports the importance of the atmospheric processing of dust for the atmospheric DP cycle as well as the potential underestimate of the DP source in all sensitivity simulations. Such underestimate could be associated with an underestimate in the primary source or in the secondary (atmospheric processing) of DP and deserves further studies.

Considering the scarcity of observational data and the gaps in knowledge of P emissions and fate in the atmosphere, the simulated atmospheric P aerosol concentrations (N=1885) satisfactorily compare with the respective available observations (NMB = -67%) for TP (N=585) and PO4 (N=1300), and for P dry deposition fluxes (N=819; NMB=-63%), indicating however an overall model underestimate of the observed values (Fig. 4b). Based on these comparisons, we evaluate that an uncertainty of about 70% is associated with PRESENT model estimates.

**3.3     Global distribution of atmospheric phosphorus**

TM4-ECPL calculates global TP and DP atmospheric burdens of 0.011 Tg-P and of 0.003 Tg-P, respectively. The calculated global annual mean TP and DP atmospheric surface distributions for the present day are also shown in Fig. 5a and Fig. 5b. TP surface concentrations maximize over the major dust regions of the world, roughly 0.1-1 $\mu$g-P m$^{-3}$ (Fig. 5a), where P-containing dust particles dominate the TP burden. Secondary maxima are calculated over Central Africa, Asia and Indonesia, where significant TP concentrations (10-100 ng-P m$^{-3}$) are associated with biomass burning emissions and PBAP (Fig. 5a). Over the oceans however, TP concentrations maximize downwind of dust source regions (roughly 10-100 ng-P m$^{-3}$) and secondary maxima of about 1-10 ng-P m$^{-3}$ are calculated due to long range transport from continental sources, mainly over the NH.

Annual mean DP concentrations of 100 ng-P m$^{-3}$ are calculated to occur over the Sahara, the Arabian and the Gobi deserts near the surface (Fig. 5b). The outflow from these source regions transports DP over the global ocean where annual mean concentrations of about 10 ng-P m$^{-3}$ are calculated downwind of dust source regions, with the highest impact calculated for the tropical Atlantic Ocean. The simulated concentrations of DP over polluted regions range from 1 to 10 ng-P m$^{-3}$, further highlighting the importance of anthropogenic contributions to the DP atmospheric burden - directly due to combustion emissions and indirectly due to the solubilisation of P when dust is mixed with atmospheric pollution during atmospheric transport (Fig. 5b). TP emissions associated with African dust are calculated to significantly affect the lower troposphere (Fig. 5c). Furthermore, DP shows non-negligible concentrations in the middle troposphere (Fig. 5d) that are attributed to transport from the source regions and to atmospheric ageing (mainly P-solubilisation processes) that converts insoluble to soluble P, as already discussed.

### 3.4    Present day phosphorus deposition flux

TM4-ECPL calculates that 1.300 Tg-P yr$^{-1}$ of TP are deposited to the Earth's surface of which about 0.281 Tg-P yr$^{-1}$ over the ocean (Table 4). This oceanic deposition flux is calculated to be about half of that estimated by Mahowald et al. (2008) (0.558 Tg-P yr$^{-1}$) over oceans and at the low end of the deposition flux range calculated by Wang et al. (2014) (0.2-1.6 Tg-P yr$^{-1}$ over the ocean). The highest TP annual deposition fluxes (up to 100 ng-P m$^{-2}$ s$^{-1}$) are calculated to occur over the Sahara and Gobi deserts while deposition fluxes up to 1 ng-P m$^{-2}$ s$^{-1}$ are also calculated at the outflow from dust source regions, especially over the Equatorial Atlantic and Northern Pacific Oceans (Fig. 6a). The computed global DP deposition is calculated equal to 0.455 Tg-P yr$^{-1}$, of which 0.169 Tg-P yr$^{-1}$ is deposited over the ocean (Table 4), that is about 75% higher than the estimate by Mahowald et al. (2008) (0.096 Tg-P yr$^{-1}$). The differences between the aforementioned studies can be explained on one hand by the P-solubilisation processes that only the present study take into account and thus a greater amount of PO4 is deposited at the Earth's surface, and on the other hand by the different aerosol size representation that impacts on the lifetime of airborne P-containing particles in the atmosphere. For this work, the highest DP deposition fluxes are simulated to occur downwind of dust source regions, owing to the DP content of the primary P emissions discussed in Sect. 3.1, and to P-solubilisation during atmospheric transport (Fig. 6b). Secondary DP deposition flux maxima (about 0.1 ng-P m$^{-2}$ s$^{-1}$) are simulated downwind of highly forested regions (i.e. Amazonia, Central Africa and Indonesia), reflecting the contribution of PBAPs to the DOP concentrations in the atmosphere.

Figure S7 further presents the seasonal variability of DP deposition fluxes as calculated by TM4-ECPL. The maximum seasonal DP deposition flux over the ocean of 0.049 Tg-P is calculated to occur during June-July-August (Fig. S7c), followed by 0.048 Tg-P during March-April-May (Fig. S7b) and by 0.038 Tg-P during September- October-November (Fig. S7d). The maximum DP deposition flux in summer occurs when ocean stratification also maximizes thus leading to the highest impact of atmospheric deposition to the marine ecosystems (Christodoulaki et al., 2013). Furthermore, PBAP contribution maximizes in summer at regions with important biogenic emissions (Figure S8e-h), while dust contribution maximizes in spring mainly over and downwind the major deserts in the tropical and mid-latitudes of the northern hemisphere (Figure S8a-d). This is because the enhanced photochemistry during NH spring and summer increases atmospheric oxidants and the atmospheric acidity due to NO$_x$ and SO$_x$ oxidation. Note also that under equinox conditions, in particular in spring, Sahara dust outbreaks are also favoured (Fig. S8b). Considering that most TP emissions occur in the NH (Fig. 2a), DP secondary formation from IP and OP sources are simulated to maximize there (Fig. 2c,d), with emissions from biomass burning and combustion of anthropogenic origin further contributing to the DP deposition flux.

### 3.5   Phosphorus solubility

The present-day P solubility of deposited aerosols (hereafter SP = %DP/TP) is calculated to vary spatially (Fig. 7a), with minima (as low as 10%) over dust source regions like the Sahara (where the insoluble fraction of TP dominates aerosol content) and maxima (up to roughly 90%) over remote oceans such as the equatorial Pacific, the southern Atlantic, the Indian and the Southern Oceans. Over such remote oceanic regions, high solubility fractions are calculated due to low P-containing aerosol mass concentrations, that occur via the long-range transport of fine particles from distance source regions, and the P which is associated with more aged aerosols and thus a greater fraction is present in the soluble mode; either as DIP via mineral acid solubilisation processes or DOP via atmospheric oxidation of P-containing organic aerosols and as PBAPs. Vet et al. (2014) in their review paper for nutrients deposition, also mentioned that the P solubility fractions of wet-only samples on coastal and inland sites have been measured to range from 30% to 90%, reflecting thus the effects of combustion, biomass burning, and phosphate fertilizers on airborne phosphorus concentrations. Anderson et al. (2010) reported that only 15-30 % of P in atmospheric aerosols at the Gulf of Aqaba was water soluble phases or relatively soluble to be bioavailable to the ecosystems. In the Mediterranean the measured median solubilities of the inorganic fraction of P in aerosols (ratio of PO4 to total inorganic P) range between 20% and 45% in the East Mediterranean with the lowest values in dust influenced air masses and the highest values in air masses from the European continent (Markaki et al., 2003; Herut et al., 1999) and have been reported to be around 38% in the West Mediterranean (Markaki et al., 2010). However, simultaneous observations of TP and DP deposition fluxes are required to evaluate the solubility fraction of P (both organic and inorganic) over remote oceans and thus to understand the atmospheric fate of P. There are only a few aerosol data available in the literature for the marine atmosphere (Graham and Duce, 1982; Baker et al., 2006a; Baker et al., 2006b; Zamora et al., 2013) that provide hints on the total P solubility. These data indicate P solubilities ranging overall between 0.01% and 94%, with the lowest values corresponding to dust influenced air masses and the highest to seasalt influenced air masses. Over the northern hemisphere Atlantic ocean P solubilities in aged Saharan dust aerosols have been measured to range from 0.01 to 37% during oceanographic cruises (Baker et al., 2006a;Baker et al., 2006b). At Barbados island median solubilities of P on dust of about 19% and of seasalt aerosol of about 94% have been reported (Zamora et al., 2013). In the southern Atlantic atmosphere P-solubilities in aerosols of up to 67% (median 8% for dust aerosol and 17% for southern Atlantic aerosol; Baker et al., 2006a) and of up to 87% (median 32%; Baker et al., 2006b) have been reported. These studies but one report P solubility as the ratio of PO4 $^-$ to TP, thus neglecting the organic fraction which has been measured to be about 28-44% (Zamora et al., 2013). Although these observations support high P solubilities in aged aerosols or aerosols impacted by non- dust sources supporting the findings of our modelling study, only the work by (Zamora et al., 2013) could be compared to the here simulated total P solubility (Fig. 7a). They indicate that the model simulated total P solubility is at the upper edge of observed P solubilities.

The soluble P originating from each source as a fraction of TP from all sources is shown in Fig. S9 for all P source categories (within each model grid these fractions sum up to 100%). Note here that in our SP fractions calculations we also include the contribution of DOP from supercoarse PBAP. This assumption is followed since the DOP from PBAP is considered readily bioavailable, compared to other super-coarse particles such as dust for which the bioavailability is characterized mainly from the initial dust's solubility rather than atmospheric processing due to the short atmospheric lifetime of super-coarse particles. Indeed, super-coarse particles of that size are basically emitted and deposited in the same model gridbox (Brahney et al. 2015), and are not therefore expected to significantly impact SP fractions over remote oceanic areas.

The low SP values over dust source regions are mainly attributed to the relatively low both weathering of dust aerosols (10%) assumed in emission fluxes and mineral P-dissolution rate (Fig. S9a). The low water associated with dust aerosols near dust sources and the enhanced buffering capacity of dust carbonate leading to excess of $Ca^{2+}$ concentrations (see Sect. 3.1 and Fig. 2c) thus cause low P dissolution. The model calculates high SP values (up to 50-60%) over regions such as the Mediterranean basin, where the co-existence of relatively high dust concentrations and high amounts of anthropogenic pollutants (e.g. Kanakidou et al., 2011) tends to enhance significantly atmospheric processing of mineral P (Nenes et al., 2011). High dust SP values are also calculated over the open ocean of the NH, the Atlantic Ocean, the Pacific Ocean in the outflow of the America, downwind of the Arabian Desert over the Indian Ocean and over the European continent. These results are attributed to the mineral P-solubilisation under polluted acidic atmospheric conditions.

Anthropogenic combustion aerosols are calculated to contribute significantly to SP (20-30%) over highly populated regions of the world, mainly over the NH as in the case of the eastern and the western coasts of the US, central and Northern Europe and Western Asia (Fig. S9b). About 5-15% of the calculated SP over the remote oceans is attributed to long range transport, where aerosols have been subjected to atmospheric ageing. Biomass burning aerosols are calculated to contribute regionally less than 30% to SP, with their maximum contribution over the equatorial Atlantic and Indian Oceans due to aerosol transport and the atmospheric ageing from Central Africa and India (Fig. S9c). DP emissions and atmospheric ageing associated with PBAPs from terrestrial sources including super-coarse P containing bioaerosols (i.e. pollen in the present study) are calculated to significantly contribute to DP deposition in the tropics; about 50% in the outflow of Amazonia and Central African and Indonesian forests on annual mean basis (Fig. S9d). Seasonally, this contribution is even higher during summer, for instance it reaches 60% in the Mediterranean and 70% in the outflow over the equatorial Pacific Ocean (not shown). DP from sea-spray dominates over all the remote Southern Ocean where no other significant primary source of DP is present (Fig. S9e), while volcanic eruptions contribute to the SP mainly over the equatorial and Northern Pacific Ocean (Fig. S9f).

## 4    Sensitivity of soluble phosphorus budget to air-pollutants

Atmospheric acidity strongly depends on anthropogenic $SO_x$, $NO_x$ and $NH_x$ emissions and impacts on dust solubility. It is thus expected to change in response to variability in the anthropogenic emissions of air pollutants (Weber et al., 2016). The response of atmospheric ageing of TP, which potentially converts the insoluble TP fraction to DP, to air pollutant emission changes is here assessed by comparing simulations performed using anthropogenic and biomass burning past and future emissions to the present-day simulation (see Sect. 2). In addition to dust dissolution changes, atmospheric OA ageing is also affected by changes in oxidants levels (Tsigaridis and Kanakidou, 2003; Tsigaridis et al., 2006). Furthermore, primary anthropogenic and biomass burning emissions of P also vary, as shown in Table 1 and discussed in Sect. 2.1. In particular, PRESENT TP anthropogenic emissions are estimated to have increased by a factor of 5 since PAST and to be reduced back to almost the PAST levels in the FUTURE. In the simulations discussed here, meteorology and natural emissions of dust, sea-salt, PBAP and from volcanoes are kept constant; to those of the year 2008 (i.e. PRESENT simulation). Although for this work we don't account for any changes in atmospheric dust emissions for PAST and FUTURE simulations, several studies suggest that dust may vary strongly and perhaps be sensitive to anthropogenic climate change and land use (Ginoux et al., 2012; Mahowald et al., 2010; Prospero and Lamb, 2003) and thus could also be an important driver of changes in the atmospheric P cycle. Overall for this study, the computed changes for species that regulate the mineral-P acid solubilisation (e.g. $SO_4^{2-}$, $NO_3^-$, $NH_4^+$) are due to the respective combustion emission differences between PAST, PRESENT and FUTURE simulations.

For the PAST simulation, the anthropogenic emissions (e.g. $NO_x$, $NH_x$ and $SO_x$) are a factor of 5-10 lower than present day emissions (Lamarque et al., 2013). Compared to the present day, the model calculates significant changes in the aerosol-pH in the past simulation with less acidic pH near the surface of the NH oceans, but a more acidic pH over the US due to extensive coal combustion in 1850 (Myriokefalitakis et al., 2015). The FUTURE simulation projects globally a less acidic aerosol pH than present day (Myriokefalitakis et al., 2015), owing to lower $NO_x$ and $SO_x$ emissions. Indeed, for the future simulation, anthropogenic emissions (RCP6.0) for most of the continental areas are projected to be lower than the present-day and to almost return to 1850 levels due to air quality regulations (Lamarque et al., 2013). However, as discussed in Myriokefalitakis et al. (2015) for the atmospheric cycle of Fe, due to the fact that biomass burning emissions are projected to increase in the future, the system does not fully return to 1850 conditions. Past and future changes of the atmospheric acidity have a significant effect on mineral-P dissolution (Fig. S10c,d) and on the ageing of atmospheric OP (Fig. S10e,f). For the PAST simulation the model calculates about 40% lower acid mineral-P dissolution (0.085 Tg-P $yr^{-1}$) compared to present-day (0.144 Tg-P $yr^{-1}$) while for the FUTURE, the acid mineral-P solubilisation (0.100 Tg-P $yr^{-1}$) is projected almost 30% lower than nowadays (Table 3).

## 4.1    Past and future changes in the phosphorus deposition flux

The global annual deposition fluxes of TP and DP as computed by TM4-ECPL for the three main simulations (PAST, PRESENT, FUTURE) are provided in Table 4. For the PAST simulation, the model calculates a global TP deposition flux (Table 4) that is about 3% lower than the present-day flux. Significant increases in TP deposition fluxes since the PAST have been calculated over Indonesia and South-eastern Asia (Fig. 6c), as a result of the present high anthropogenic emissions over China. As for the FUTURE simulation, TP deposition is projected to decrease globally by 2.5% compared to present-day (Table 4) with a maximum decrease (up to 40%) due to emission control measures calculated for Europe, the Eastern US and China.

On global scale, DP deposition fluxes are also calculated to be lower by about 20% in the PAST and by about 15% in the FUTURE simulations compared to the PRESENT one (Table 4). Although these reductions are computed to be relatively low, regional reductions can be stronger (up to 60%, Fig. 6d), especially over highly populated regions (e.g. China, Europe) and downwind of major dust sources (e.g. India and Western US). Indeed, present-day DP emission fluxes from anthropogenic combustion (0.021 Tg-P $yr^{-1}$) are calculated to decrease by about 80% on a global scale both for the PAST and FUTURE simulations (Table 1). According to our calculations, however, present-day DP biomass burning emissions have increased by about 28% from PAST and are expected to further increase by about 22% in the FUTURE. When accounting for both DP anthropogenic combustion and biomass burning emissions, an increase of about 3 times is computed from the PAST to PRESENT but a decrease to half is expected for the FUTURE, contributing thus significantly to the DP atmospheric deposition changes. Hence, DP deposition fluxes are projected to decrease over the mid-latitudes of the NH where human activities dominate (Fig. 6f), with the largest changes up to 60% over China due to the expected air-quality measures, while smaller changes are computed over India due to the expected increase in its population. Note again that our simulations neglect any change in dust and PBAPs emissions that has occurred in the past or is expected to take place in the future. Therefore, the changes in BP deposition fluxes shown in Table 4 are driven by changes in the anthropogenic and biomass burning emissions and in the atmospheric oxidants that enhance P dissolution during atmospheric ageing.

## 4.2    Biogeochemical implications of changes in bioavailable phosphorus deposition

The contribution of dust to the bioavailable P deposition flux into the ocean maximizes in the outflow from desert regions, mainly in the north hemisphere tropics and mid latitudes (Fig. S8a-d). However, according to our simulations, DOP is an important fraction of bioavailable P, mainly over continental regions. Table 4 also presents the sum of the DP and the insoluble PBAP deposition, reported as bioavailable P (BP), which is considered to be readily bioavailable for marine ecosystems since it is biological material. Figures S8e-h depict the seasonal variability of the PBAPs deposition flux computed by our model for the PRESENT simulation. It is remarkable that our simulations suggest that bioaerosols are a major contributor to the BP deposition fluxes; on an annual basis, PBAPs contribute about 25% to the global BP deposition fluxes over the oceans (about 33% on a global scale), but regionally more than 50% (Fig. 7b) in the outflow of South America over the equatorial Pacific and in the outflow of Central Africa over the Southern Tropical Atlantic. This finding clearly shows that biological material is a major atmospheric carrier of bioavailable P to the global ocean (Fig. 7b) and implies a potentially important impact of terrestrial sources on marine ecosystems. Note that as mentioned in Sect. 2, PBAPs from insect fragments and plant debris are neglected in the present study; thus their contribution to the bioavailable P deposition mainly over land might be here underestimated. However, large uncertainties are associated with this innovative finding, since the estimates of the global source of PBAPs vary by more than an order of magnitude, their size distributions, their mass density are uncertain and the P-content in these aerosols is also highly variable, spanning 2 orders of magnitude (e.g. Kanakidou et al. (2012) supplementary material and references therein). All these parameters have to be studied by targeted experiments to improve knowledge of their contribution to the atmospheric P cycle. Our results also indicate that primary anthropogenic emissions of DP, as well as anthropogenically-driven atmospheric acidity, increased the DP supply to the global ocean since the preindustrial period thus providing an important external to the ocean source of nutrients for the marine ecosystem. They also show that the P solubilisation from dust aerosol during atmospheric transport and mixing with acidic pollutants is important for DP deposition and deserves further kinetic studies to improve parameterisation of the solubilisation kinetics of various P containing minerals as a function of acidity and temperature. These results may be particularly important for ecosystems like the East Mediterranean where phytoplankton growth is limited by P availability.

It is also noteworthy that the bioavailable P deposition flux from bioaerosols maximizes in summer (Fig. S8e-h) when ocean stratification is also the strongest, thus leading to the highest impact of atmospheric deposition to the marine ecosystems (Christodoulaki et al., 2013). This flux needs to be taken into account to evaluate the atmospheric DP deposition impact on marine ecosystems. The computed atmospheric deposition of BP over the global ocean of 0.17 Tg-P yr$^{-1}$ (Table 4) represents about 15 % of the global riverine flux to the ocean of 0.99 Tg-P yr$^{-1}$ (Meybeck, 1982). However, while riverine inputs affect mainly the coastal regions, atmospheric deposition is a source of nutrients for the open sea (e.g. Okin et al., 2011).

## 5    Conclusions

In this study the global atmospheric cycle of Phosphorus has been simulated with the state-of-the-art atmospheric chemistry transport global model TM4-ECPL. The novel aspect of this study is the simultaneous consideration of primary TP and DP emissions accounting for both inorganic and organic P and of the atmospheric processing of P. Accounting for a DP (both inorganic and organic) primary source of 0.272 Tg-P yr$^{-1}$ together with a PO4 acid-solubilisation flux of 0.144 Tg-P yr$^{-1}$ and a DOP ageing flux of 0.033 Tg-P yr$^{-1}$, result overall in a present-day atmospheric DP burden of 0.003 Tg-P and a global DP annual deposition flux of 0.455 Tg-P yr$^{-1}$, of which 0.169 Tg-P yr$^{-1}$ is deposited over the oceans. P solubility in deposited aerosols is calculated to vary spatially with minima over the dust sources (<10%) and maxima over the remote ocean (up to 90%).

Sensitivity simulations show that increases in anthropogenic and biomass burning emissions since preindustrial times resulted in both enhanced DP combustion primary emissions and P-dissolution occurring under a more acidic environment. Air-quality regulations however, are projected to decrease anthropogenic emissions, mitigate oxidant levels and limit future atmospheric acidity. Focusing on oceanic regions, atmospheric composition change over the last 150 years is calculated to have increased the DP deposition to the ocean by about 30% (i.e. 0.132 Tg-P yr$^{-1}$ in preindustrial times against 0.169 Tg-P yr$^{-1}$ nowadays). Projection based on future combustion emissions, drives the model to a 30% reduction in mineral P dissolution flux (0.100 Tg-P yr$^{-1}$ in the future compared to 0.144 Tg-P yr$^{-1}$ in the present day) and taking into account an 80% reduction of the anthropogenic DP emissions, the model calculates an oceanic DP deposition flux of 0.142 Tg-P yr$^{-1}$ that is about 16% lower than present-day. Our results further indicate a significant contribution to the calculated DP deposition fluxes of DIP up to 90%, over the Northern tropical Atlantic, Pacific and Southern Oceans, as well as a DOP contribution higher than 50% over the equatorial oceanic regions.

The contribution of PBAPs deposition to the total bioavailable P to the marine environment is found to exceed 50% regionally, indicating the existence of potentially important interactions between the terrestrial and the marine biospheres. Therefore, our results provide new insights to the atmospheric P cycle by demonstrating that PBAPs are as important carriers of bioavailable P as dust aerosol, that was up to now considered as the only large source of DP external to the open ocean.

Although the present global modelling study is based on the current understanding of the processes that drive the atmospheric cycle of P, comparison of model results to observations showed that the model underestimates the data by at least 60%. Improvements, thus, require reduction in the large uncertainties that still exist with regard to the primary TP and DP emissions from anthropogenic and natural sources and the adopted kinetic parameters of mineral-P dissolution and organic aerosol-P ageing and their response to the changes in atmospheric acidity. Finally, in view of the importance of P as a nutrient for marine and terrestrial ecosystems in terms of carbon storage and nitrogen fixation, the calculated changes in P deposition due to projected air-quality changes, indicate the necessity to account for feedbacks between atmospheric chemistry, climate and biogeochemical cycles.

*Acknowledgements.* This research has been co-financed by the European Union (European Social Fund – ESF) and Greek national funds through the Operational Program "Education and Lifelong Learning" of the National Strategic Reference Framework (NSRF) - Research Funding Program: ARISTEIA – PANOPLY (Pollution Alters Natural Aerosol Composition: implications for Ocean Productivity, cLimate and air qualitY) grant. SM acknowledges the European FP7 collaborative project BACCHUS (Impact of Biogenic versus Anthropogenic emissions on Clouds and Climate: towards a Holistic UnderStanding) for financial support. The authors thank Dr. T. van Noije for dust emission module availability, Dr. R. Wang for providing anthropogenic and natural combustion P emission data, P. Nicolaou, A. Mitsotaki, Dr. C. Theodosi, Dr. P. Zarbas and Dr. G. Kouvarakis for the observations in the Mediterranean and Dr. R. Vet for providing P deposition data. The authors thank two anonymous referees for constructive comments and Dr. S. Bikkina for interactive comments on the manuscript.

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

Table Captions

Table 1. Emissions of TP and DP (in Tg-P yr$^{-1}$) taken into account in the TM4-ECPL model for PAST, PRESENT and FUTURE simulations. In parenthesis the average values of TP and DP emissions for the years 2000-2010 are also provided.

Table 2. Fluorapatite (FAP) acid dissolution constants used for this study.

Table 3. Secondary DP sources (in Tg-P yr$^{-1}$) due to OP ageing contained in biomass burning, anthropogenic combustion, sea-spray and dust as well as due to dust (apatite) dissolution via the acid solubilisation mechanism, as calculated by the TM4-ECPL model for PAST, PRESENT and FUTURE simulations.

Table 4. Global and Oceanic deposition fluxes of TP, DP and BP (in Tg-P yr$^{-1}$), as calculated by the TM4-ECPL model for PAST, PRESENT and FUTURE simulations.

Figure Captions

Figure 1. Simplified illustration of the atmospheric P-cycle showing the various sources of particulate inorganic and organic P (IPP, OPP) and their soluble forms (DIP and DOP), the transformation of PP to DP during atmospheric transport and the deposition of P to the land and to the ocean. Emissions fractions among atmospheric P forms are those used as input in the TM4-ECPL model.

Figure 2. Annual averaged column distributions (in ng-P m$^{-2}$ s$^{-1}$) of the a) TP emissions, b) DP emissions, c) DIP flux from apatite dissolution and d) DOP production due to OP atmospheric ageing, as calculated by the TM4-ECPL model for the present atmosphere (year 2008).

Figure 3. DIP annual fluxes (in ng-P m$^{-2}$ s$^{-1}$) from apatite dissolution a) in aerosol water and b) in cloud droplets, as calculated by the TM4-ECPL model for the present atmosphere (year 2008).

Figure 4. Location of observational date for a) Concentrations of P-containing aerosols (bulk, fine and coarse) and b) Deposition fluxes (wet and dry deposition); (c-f) Log-scatter plots between model (y-axis) and all observations (x-axis), for surface c) PO4 and e) TP aerosol concentrations (μg-P m$^{-3}$) measured in cruises (blue dots) and stations (red stars), as well as for d) PO4 and f) TP dry deposition fluxes (mg-P m$^{-2}$ s$^{-1}$) over oceans (blue dots) and inland sites (red stars). The continuous black line shows the 1:1 correlation and the dashed black lines show the 10:1 and 1:10 relationships, respectively.

Figure 5. Annual mean concentrations (in ng-P m$^{-3}$) of TP (a, c) and DP (b, d) for the surface (a, b) and in the troposphere as zonal mean (c, d), as calculated by the TM4-ECPL model for the present atmosphere (year 2008).

Figure 6. Calculated annual deposition fluxes (in ng-P m$^{-2}$ s$^{-1}$) for a) TP and b) DP for PRESENT simulation and their percentage differences from PAST (c, d) and FUTURE (e, f) simulations, respectively. For the PRESENT annual deposition fluxes (a, b), within brackets the total amounts over the globe (in parentheses only over ocean) are also provided.

Figure 7. Annual mean percentage fractions of a) P solubility (SP = %DP/TP) and b) the relative contribution of PBAP to BP, in deposited P-containing aerosols, as calculated by the TM4-ECPL model for the present atmosphere (year 2008).

**Table 1. Emissions of TP and DP (in Tg-P yr$^{-1}$) taken into account in the TM4-ECPL model for PAST, PRESENT and FUTURE simulations. In parenthesis the average values of TP and DP emissions for the years 2000-2010 are also provided.**

| | TM4-ECPL | Biomass Burning | Anthropogenic Combustion | Volcanoes | PBAP | Sea Spray | Soils | Total |
|---|---|---|---|---|---|---|---|---|
| **TP** | **PAST** | 0.014 | 0.008 | | | | | 1.289 |
| | **PRESENT** | 0.018 | 0.043 | 0.006 | 0.156 | 0.008 | 1.097 | 1.328 |
| | | (0.018) | (0.042) | (0.006) | (0.156) | (0.008) | (1.095) | (1.326) |
| | **FUTURE** | 0.022 | 0.009 | | | | | 1.298 |
| **DP** | **PAST** | 0.007 | 0.004 | | | | | 0.254 |
| | **PRESENT** | 0.009 | 0.021 | 0.006 | 0.123 | 0.008 | 0.106 | 0.272 |
| | | (0.009) | (0.021) | (0.006) | (0.123) | (0.007) | (0.105) | (0.271) |
| | **FUTURE** | 0.011 | 0.004 | | | | | 0.258 |

5  **Table 2. Fluorapatite (FAP) acid dissolution constants used for this study.**

| Mineral* | pH | K(T) (mol m$^{-2}$ s$^{-1}$) | m | A$_{MIN}$ (m$^2$ g$^{-1}$) | K$_{eq}$ |
|---|---|---|---|---|---|
| **FAP** | <5.5 | 5.75x10$^{-6}$exp[4.1x10$^3$(1/298-1/T)] [a] | 0.81 [a] | 10.7 [b] (FAP) | 10$^{-23.12}$ [d] (FAP) |
| | 5.5 – 6.5 | 6.91x10$^{-8}$exp[4.1x10$^3$(1/298-1/T)] [a] | 0.67 [a] | | |
| | >6.5 | 6.53x10$^{-11}$exp[4.1x10$^3$(1/298-1/T)] [a] | 0.01 [a] | 80.5 [c] (HAP) | 10$^{-20.47}$ [d] (HAP) |

*a) Guidry and Mackenzie (2003); b) Bengtsson et al. (2007); c) Bengtsson et al. (2009); d) van Cappellen and Berner (1991)*

*For HAP dissolution constants, we assume those of FAP as adopted from Guidry and Mackenzie (2003) and corrected based on Palandri and Kharaka (2004) reviewed data (see Sect. 2.2)*

10  **Table 3. Secondary DP sources (in Tg-P yr$^{-1}$) due to OP ageing contained in biomass burning, anthropogenic combustion, sea-spray and dust as well as due to dust (apatite) dissolution via the acid solubilisation mechanism, as calculated by the TM4-ECPL model for PAST, PRESENT and FUTURE simulations.**

| 6 | TM4-ECPL | Biomass Burning Ageing | Anthropogenic Combustion Ageing | PBAP Ageing | Sea Spray Ageing | Dust Ageing | |
|---|---|---|---|---|---|---|---|
| | | | | | | *OP Ageing* | *Apatite Dissolution* |
| **DP** | **PAST** | 0.002 | 0.001 | 0.016 | 0.0001 | 0.007 | 0.085 |
| | **PRESENT** | 0.003 | 0.005 | 0.016 | 0.0001 | 0.008 | 0.144 |
| | **FUTURE** | 0.003 | 0.001 | 0.016 | 0.0001 | 0.008 | 0.100 |

**Table 4. Global and Oceanic deposition fluxes of TP, DP and BP (in Tg-P yr$^{-1}$), as calculated by the TM4-ECPL model for PAST,**
15  **PRESENT and FUTURE simulations.**

| Deposition | TM4-ECPL | Global | Ocean |
|---|---|---|---|
| **TP** | **PAST** | 1.262 | 0.270 |
| | **PRESENT** | 1.300 | 0.281 |
| | **FUTURE** | 1.270 | 0.272 |
| **DP** | **PAST** | 0.369 | 0.133 |
| | **PRESENT** | 0.455 | 0.169 |
| | **FUTURE** | 0.390 | 0.142 |
| **BP** | **PAST** | 0.384 | 0.135 |
| | **PRESENT** | 0.470 | 0.172 |
| | **FUTURE** | 0.405 | 0.144 |

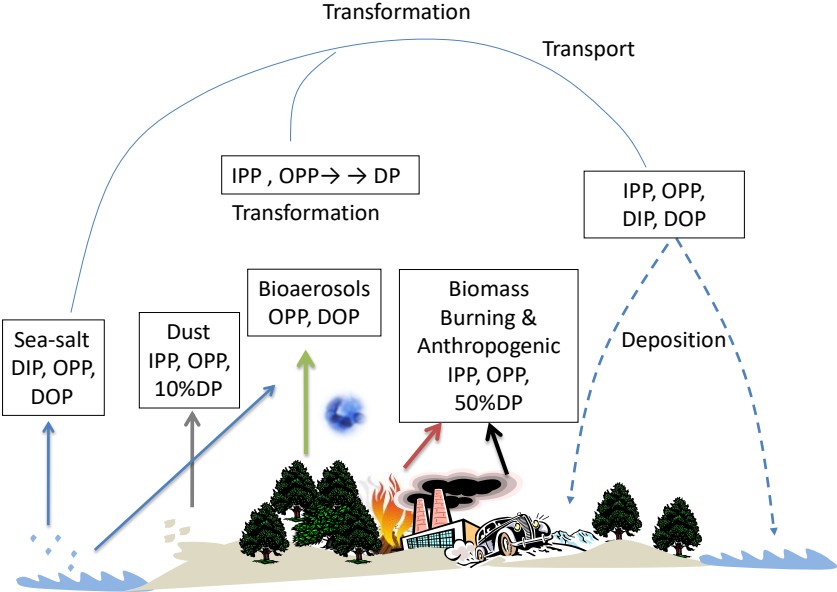

**Figure 1. Simplified illustration of the atmospheric P-cycle showing the various sources of particulate inorganic and organic P (IPP, OPP) and their soluble forms (DIP and DOP), the transformation of PP to DP during atmospheric transport and the deposition of P to the land and to the ocean. Emissions fractions among atmospheric P forms are those used as input in the TM4-ECPL model.**

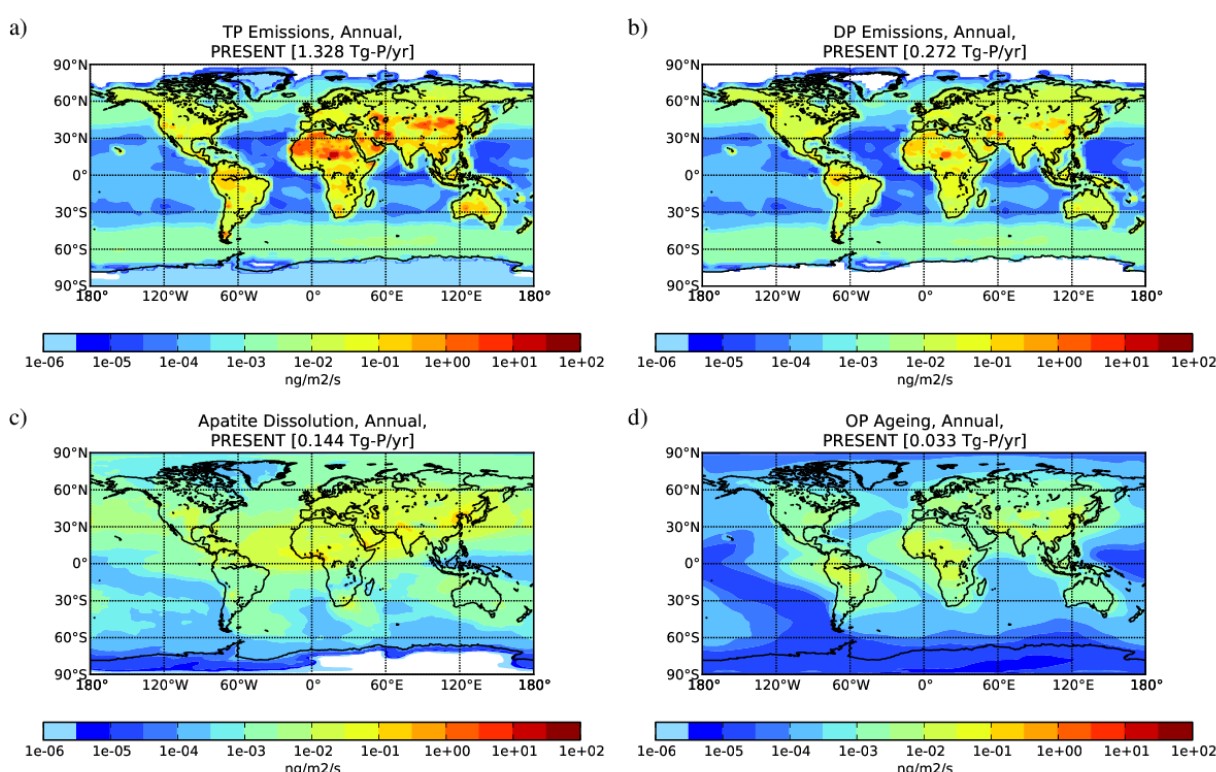

**Figure 2. Annual averaged column distributions (in ng-P m$^{-2}$ s$^{-1}$) of the a) TP emissions, b) DP emissions, c) DIP flux from apatite dissolution and d) DOP production due to OP atmospheric ageing, as calculated by the TM4-ECPL model for the present atmosphere (year 2008).**

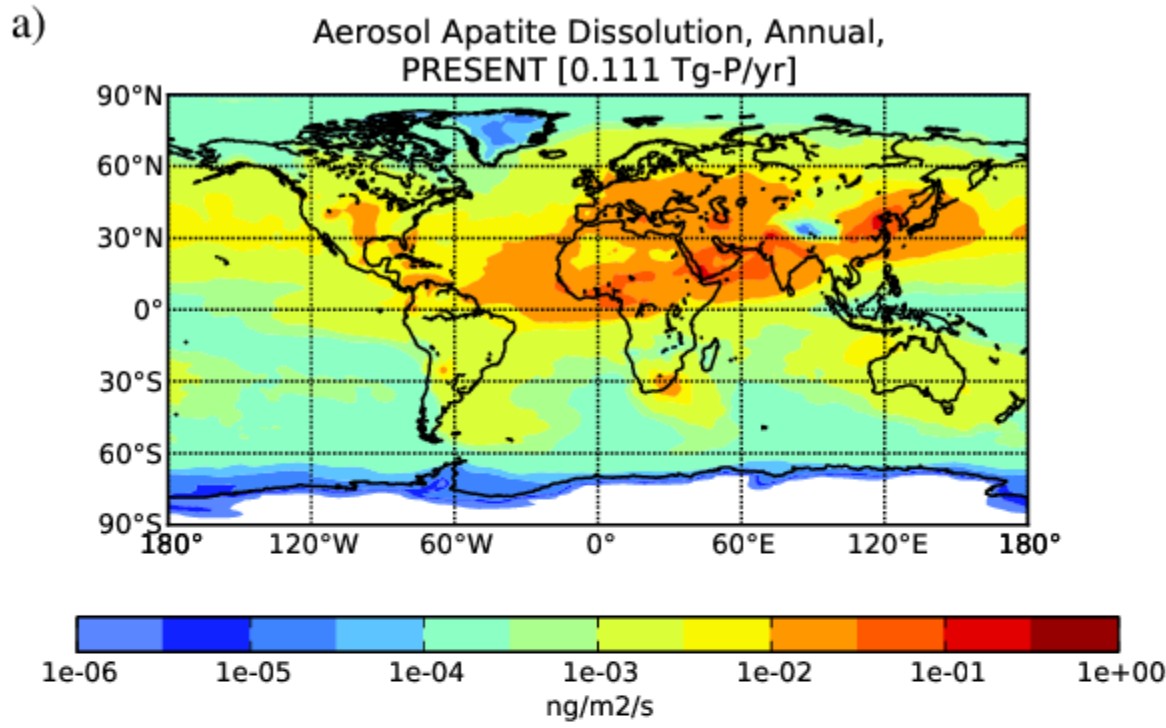

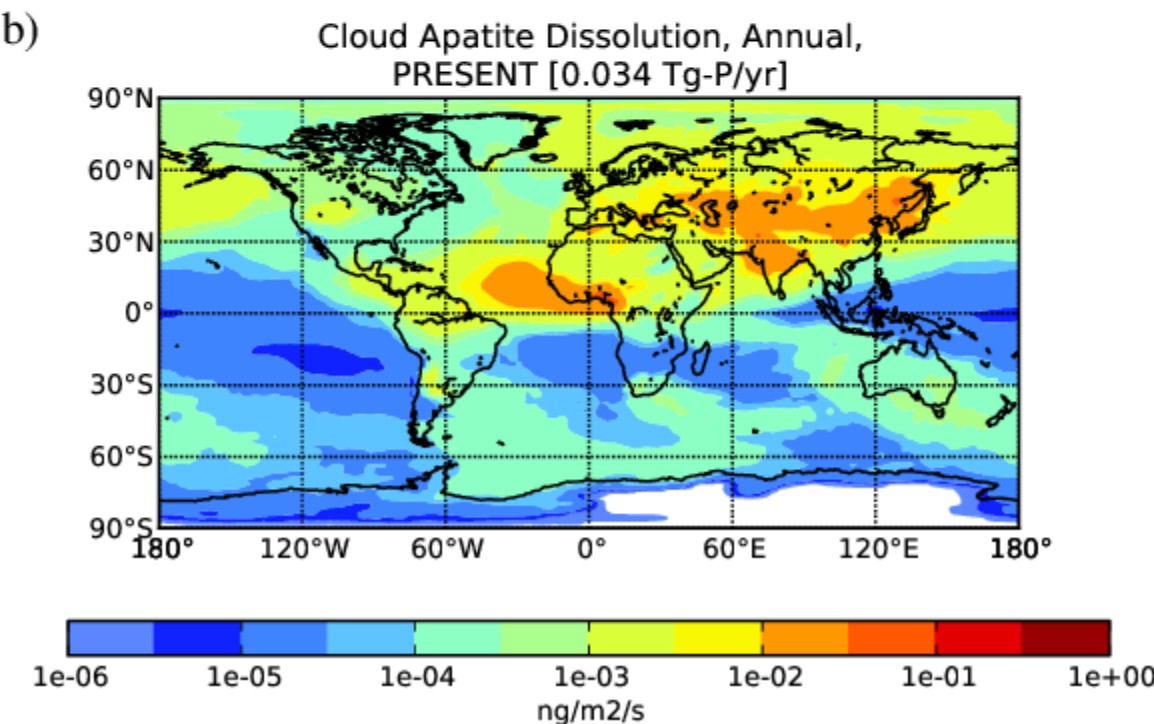

**Figure 3. DIP annual fluxes (in ng-P m$^{-2}$ s$^{-1}$) from apatite dissolution a) in aerosol water and b) in cloud droplets, as calculated by the TM4-ECPL model for the present atmosphere (year 2008).**

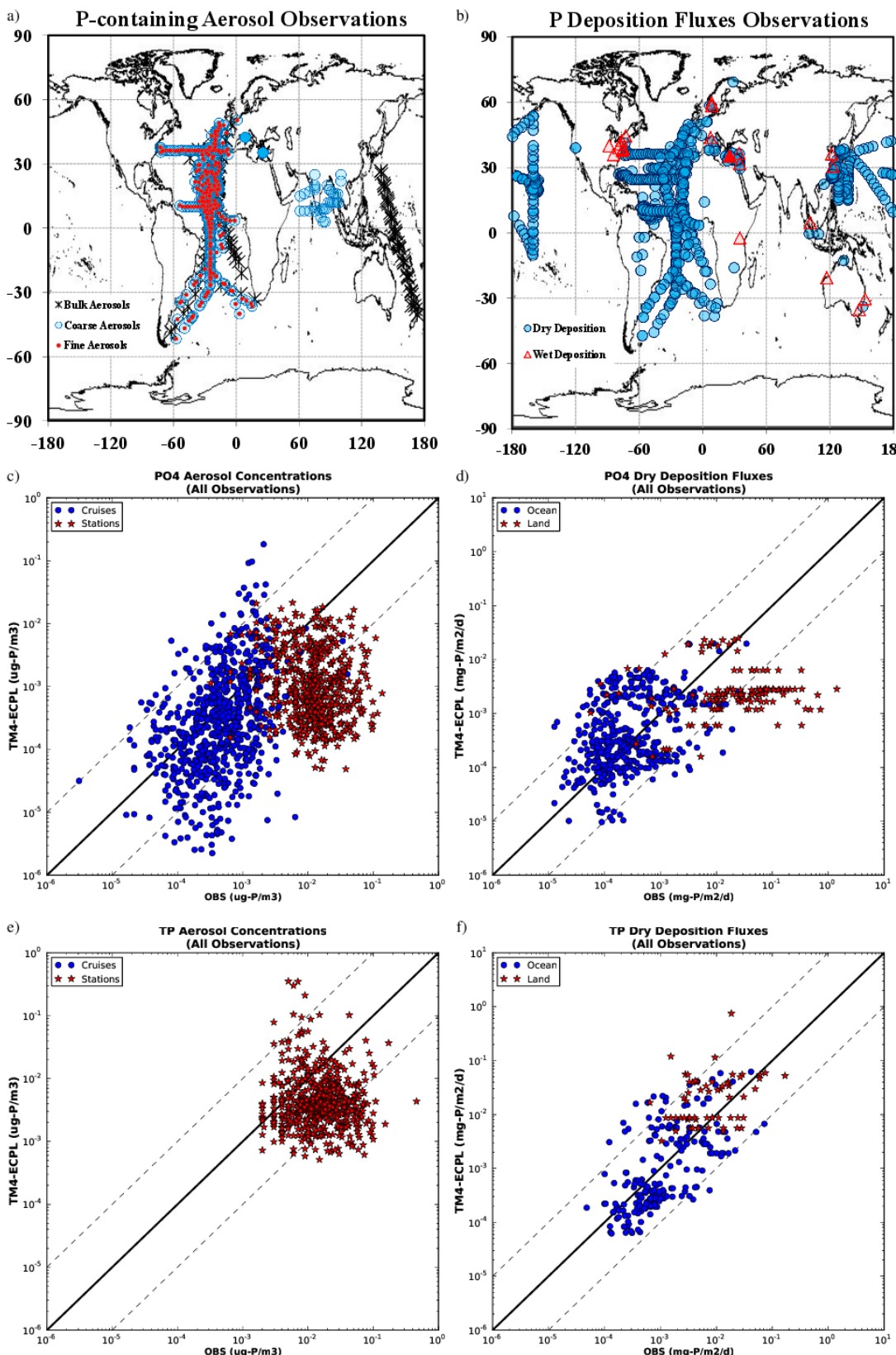

**Figure 4. Location of observational date for a) Concentrations of P-containing aerosols (bulk, fine and coarse) and b) Deposition fluxes (wet and dry deposition); (c-f) Log-scatter plots between model (y-axis) and all observations (x-axis), for surface c) PO4 and e) TP aerosol concentrations (µg-P m⁻³) measured in cruises (blue dots) and stations (red stars), as well as for d) PO4 and f) TP dry deposition fluxes (mg-P m⁻² s⁻¹) over oceans (blue dots) and inland sites (red stars). The continuous black line shows the 1:1 correlation and the dashed black lines show the 10:1 and 1:10 relationships, respectively.**

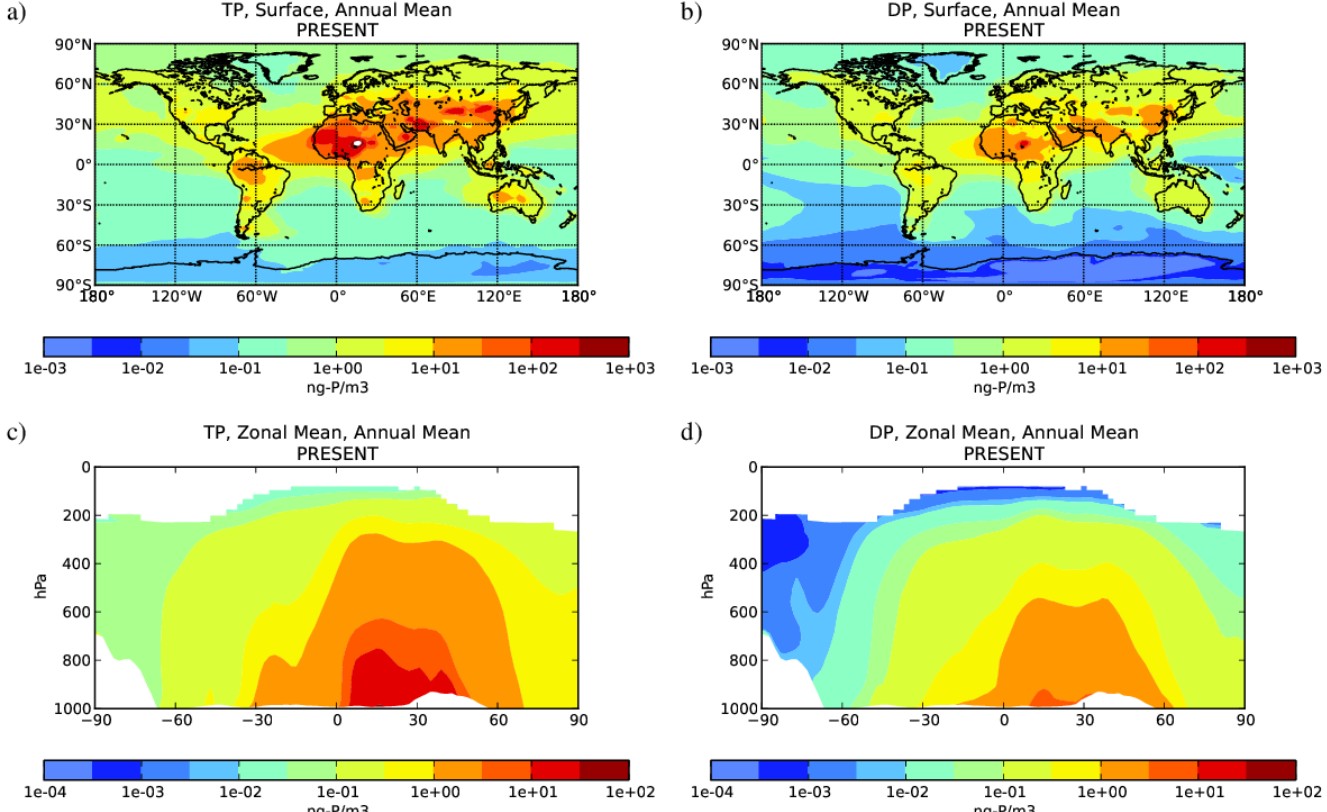

**Figure 5. Annual mean concentrations (in ng-P m$^{-3}$) of TP (a, c) and DP (b, d) for the surface (a, b) and in the troposphere as zonal mean (c, d), as calculated by the TM4-ECPL model for the present atmosphere (year 2008).**

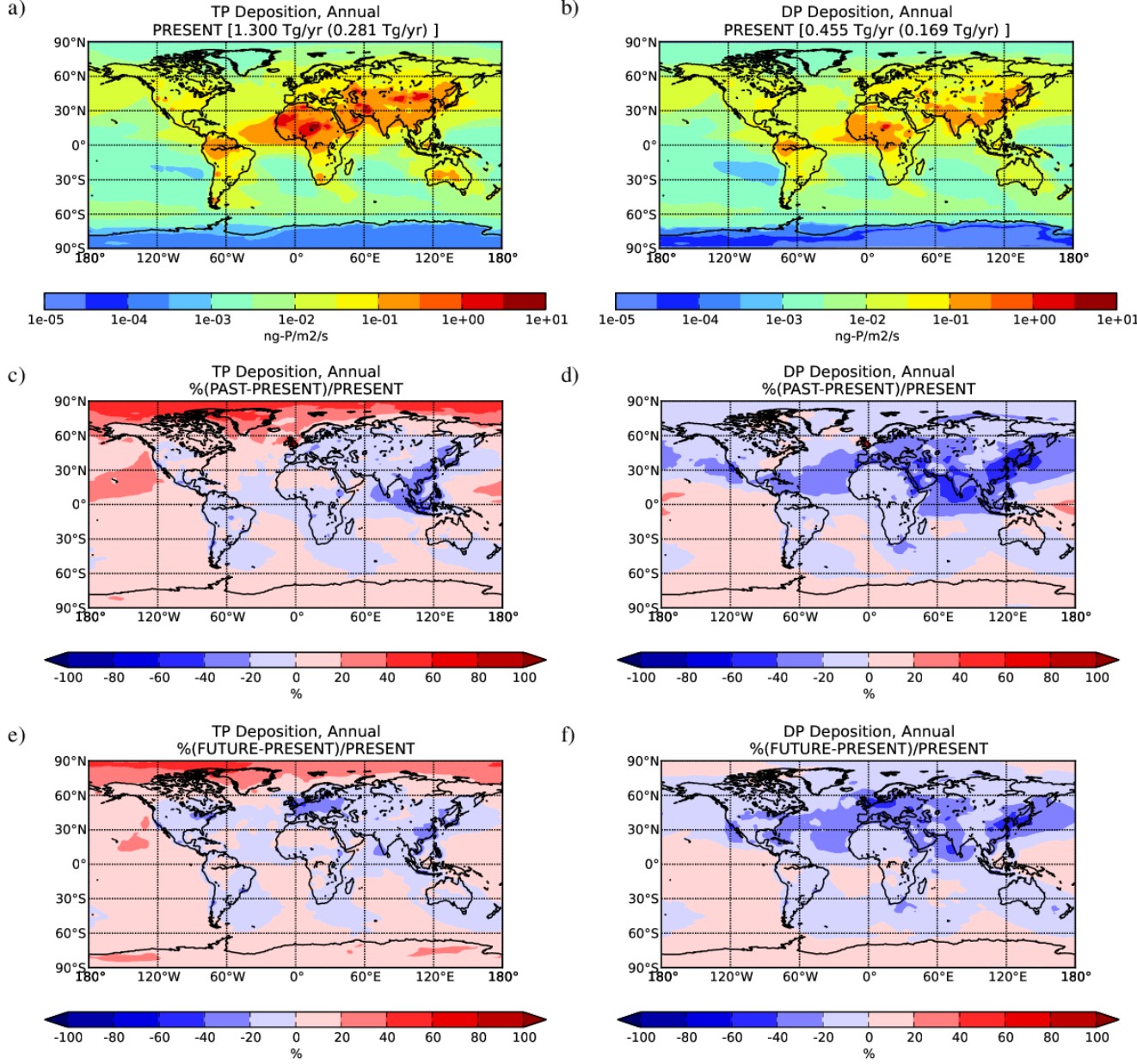

**Figure 6. Calculated annual deposition fluxes (in ng-P m$^{-2}$ s$^{-1}$) for a) TP and b) DP for PRESENT simulation and their percentage differences from PAST (c, d) and FUTURE (e, f) simulations, respectively. For the PRESENT annual deposition fluxes (a, b), within brackets the total amounts over the globe (in parentheses only over ocean) are also provided.**

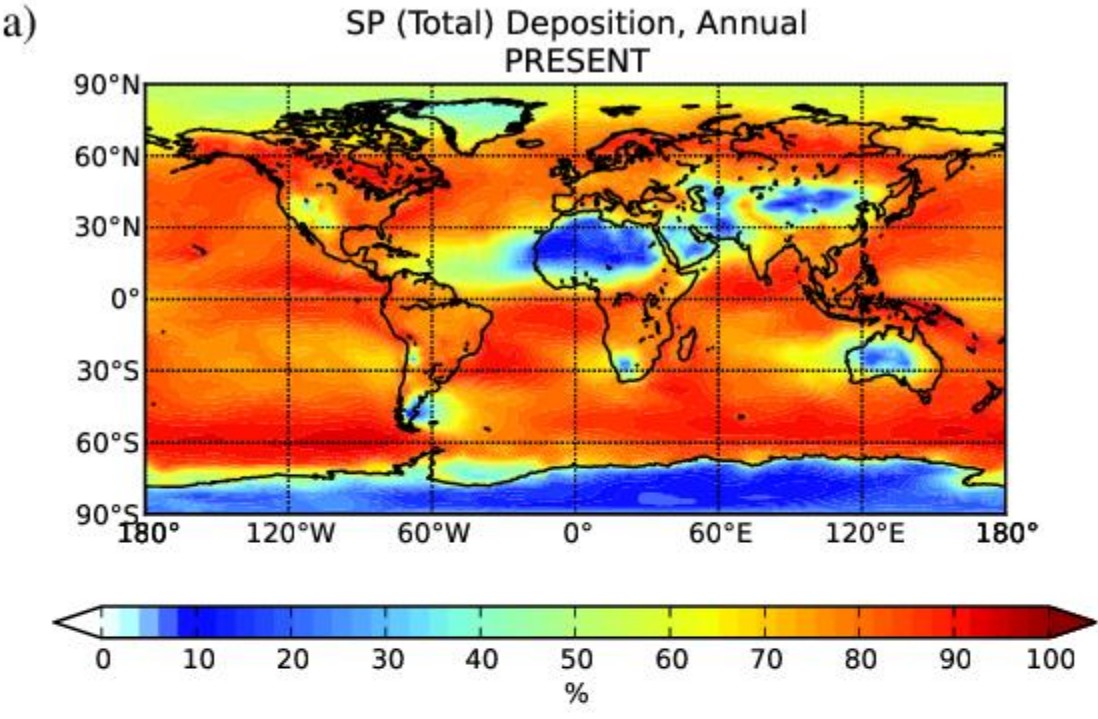

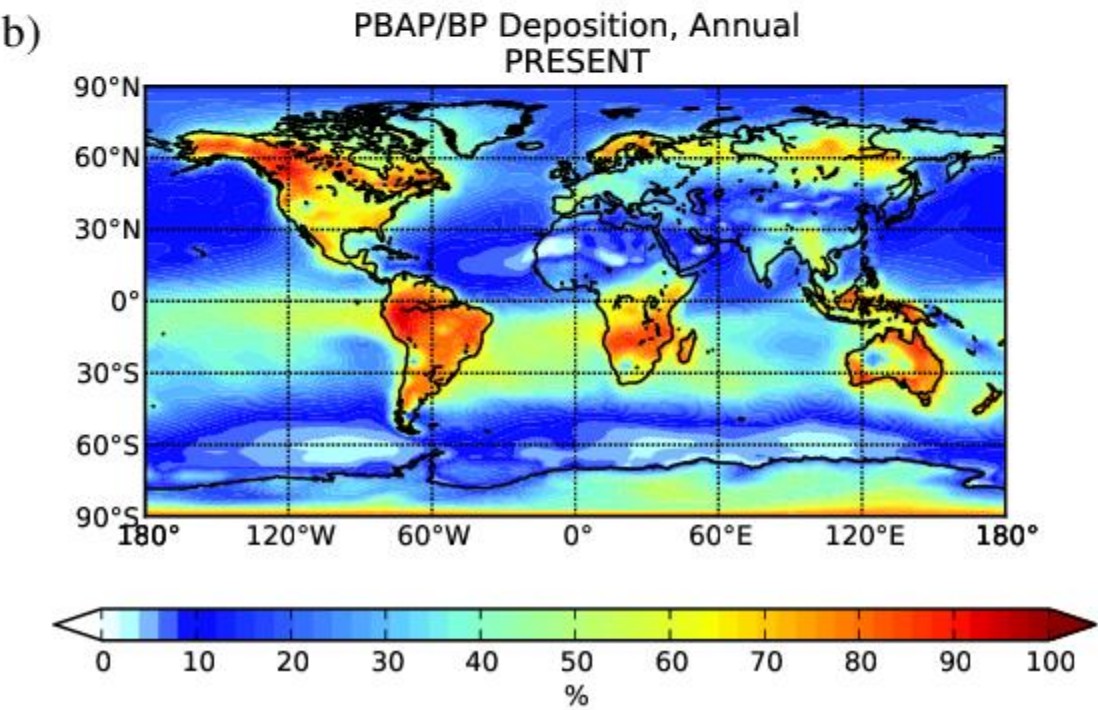

**Figure 7. Annual mean percentage fractions of a) P solubility (SP = %DP/TP) and b) the relative contribution of PBAP to BP, in deposited P-containing aerosols, as calculated by the TM4-ECPL model for the present atmosphere (year 2008).**