# Peer review of "Bioavailable atmospheric phosphorous supply to the global ocean: a 3-D global modelling study"

_Biogeosciences, 2016_

## Short Comment (SC1) · 9 Jun 2016

Comments from Srinivas Bikkina Manuscript: Bioavailable atmospheric phosphorous supply to the global ocean: a 3-D global modelling study by Myriokefalitakis et al., (2016).

This article reviews the current state of knowledge on the total and dissolved phosphorous emissions to the atmosphere and their relevance to biogeochemical impact on carbon by providing more bioavailable phosphorous through air-sea deposition. The article also highlights the need for considering the impact of bioavailable phosphorous emissions from anthropogenic sources and acid processing of mineral dust in a changing climate scenario. Authors have reviewed the existing field based and model predicted atmospheric PInorg (dissolved pool of inorganic phosphorous) deposition over the Atlantic and Pacific Oceans. However, no information is provided on the available data and observations made over the Northern Indian Ocean (Arabian Sea and Bay of Bengal) [Srinivas and Sarin, 2012; Srinivas and Sarin, 2013; Srinivas and Sarin, 2015].

Studies from Northern Indian Ocean have reported an order of magnitude difference in the atmospheric abundance of PInorg between Bay of Bengal (1.1±0.3 nmol m-3) and the Arabian Sea (0.3±0.1 nmol m-3) [Srinivas and Sarin, 2012]. Air mass back trajectories and characteristic elemental ratios of Ca/Al and Fe/Al (an index of mineral dust) suggest the source of PInorg from Arabian, Iranian and Thar Deserts. A significant ($p < 0.05$) linear relationship of PInorg with nss-Ca2+ and nss-SO42- over the Bay of Bengal provides evidence for acid processing of mineral dust (dissolution of apatite). This linear relationship between PInorg and nss-Ca2+ is not significant over the Arabian Sea [Srinivas and Sarin, 2012].

Several studies have documented dominance of anthropogenic sources over the BoB [Kumar et al., 2008; Srinivas et al., 2011]. We also computed dust derived aerosol phosphorous (Pdust, estimated by assuming P/Al ratio in the upper continental crust (0.008) and measured Al concentration and anthropogenic water-soluble inorganic phosphorous (Panth = PInorg – Pdust) contribution over the Northern Indian Ocean. A comparison of air-sea deposition of PInorg over the Northern Indian Ocean (0.035 Tg-P yr-1) is of comparable magnitude with the riverine supply [Srinivas and Sarin, 2013] and model based deposition flux (0.045 Tg-P yr-1) by Okin et al. [2011]. These calculations suggest predominance of anthropogenic sources (biomass burning emissions and fertilizers) over the Bay of Bengal (Panth: ∼75%) and contribution from mineral dust over the Arabian Sea (Pdust: ∼70%) [Srinivas and Sarin, 2012]. These results highlight the importance of atmospheric source in influencing the biogeochemical cycle of phosphorus in the Northern Indian Ocean.

The article by Myriokefalitakis et al., (2016) published as discussion paper in Biogeosciences may consider these results from the northern Indian Ocean.

**References**

Kumar, A., A. K. Sudheer, and M. M. Sarin (2008), Chemical characteristics of aerosols in MABL of Bay of Bengal and Arabian Sea during spring inter-monsoon: A comparative study, Journal of Earth System Science, 117(1), 325-332, doi:10.1007/s12040-008-0035-9. Okin, G. S., et al. (2011), Impacts of atmospheric nutrient deposition on marine productivity: Roles of nitrogen, phosphorus, and iron, Global Biogeochemical Cycles, 25(2), n/a-n/a, doi:10.1029/2010GB003858. Srinivas, B., and M. Sarin (2012), Atmospheric pathways of phosphorous to the Bay of Bengal: contribution from anthropogenic sources and mineral dust, Tellus B, 64, doi.org/10.3402/tellusb.v3464i3400.17174. Srinivas, B., and M. Sarin (2015), Atmospheric deposition of phosphorus to the Northern Indian Ocean, CURRENT SCIENCE, 108(7), 1300. Srinivas, B., and M. M. Sarin (2013), Atmospheric deposition of N, P and Fe to the Northern Indian Ocean: Implications to C- and N-fixation, Science of The Total Environment, 456–457, 104-114, doi:http://dx.doi.org/10.1016/j.scitotenv.2013.03.068. Srinivas, B., M. M. Sarin, and V. V. S. S. Sarma (2011), Atmospheric dry deposition of inorganic and organic nitrogen to the Bay of Bengal: Impact of continental outflow, Marine Chemistry, 127(1–4), 170-179, doi:http://dx.doi.org/10.1016/j.marchem.2011.09.002.

Please also note the supplement to this comment:
http://www.biogeosciences-discuss.net/bg-2016-215/bg-2016-215-SC1-supplement.pdf

**Supplement:**

**Comments from Srinivas Bikkina**

Manuscript: Bioavailable atmospheric phosphorous supply to the global ocean: a 3-D global modelling study by Myriokefalitakis et al., (2016).

This article reviews the current state of knowledge on the total and dissolved phosphorous emissions to the atmosphere and their relevance to biogeochemical impact on carbon by providing more bioavailable phosphorous through air-sea deposition. The article also highlights the need for considering the impact of bioavailable phosphorous emissions from anthropogenic sources and acid processing of mineral dust in a changing climate scenario. Authors have reviewed the existing field based and model predicted atmospheric $P_{Inorg}$ (dissolved pool of inorganic phosphorous) deposition over the Atlantic and Pacific Oceans. However, no information is provided on the available data and observations made over the Northern Indian Ocean (Arabian Sea and Bay of Bengal) (References: Srinivas and Sarin, Tellus B, 2012; Srinivas and Sarin, SOTENV, 2013; Srinivas and Sarin, Current Science, 2015).

Studies from Northern Indian Ocean have reported an order of magnitude difference in the atmospheric abundance of $P_{Inorg}$ between Bay of Bengal (range: 20-35 nmol $m^{-3}$) and the Arabian Sea (range: 0.02-0.1 nmol $m^{-3}$) (Srinivas and Sarin, 2012). Air mass back trajectories (AMBTs) and characteristic elemental ratios of Ca/Al and Fe/Al (an index of mineral dust) suggest source of $P_{Inorg}$ from Arabian, Iranian and Thar Deserts. A significant ($p < 0.05$) linear relationship of $P_{Inorg}$ with $nss\text{-}Ca^{2+}$ and $nss\text{-}SO_4^{2-}$ over the Bay of Bengal provides evidence for acid processing of mineral dust (dissolution of apatite). This linear relationship between $P_{Inorg}$ and $nss\text{-}Ca^{2+}$ is not significant over the Arabian Sea (Srinivas and Sarin, 2012).

Several studies have documented dominance of anthropogenic sources over the BoB (Kumar et al., 2008; Srinivas et al., 2011). We also computed dust derived aerosol phosphorous ($P_{dust}$, estimated by assuming P/Al ratio in the upper continental crust (0.008) and measured Al concentration and anthropogenic water-soluble inorganic phosphorous ($P_{anth} = P_{Inorg} - P_{dust}$) contribution over the Northern Indian Ocean. A comparison of air-sea deposition of $P_{Inorg}$ over the Northern Indian Ocean (0.035 Tg-P $yr^{-1}$) is of comparable magnitude with the riverine supply (Srinivas and Sarin, 2013) and model based deposition flux (0.045 Tg-P $yr^{-1}$) by Okin et al. (2011). These calculations suggest predominance of anthropogenic sources (BBEs and fertilizers) over the Bay of Bengal ($P_{anth}$: ~75%) and contribution from mineral dust over the Arabian Sea ($P_{dust}$: ~70%) (Srinivas and Sarin, 2012). These results highlight the importance of atmospheric source in influencing the biogeochemical cycle of phosphorus in the Northern Indian Ocean.

The article by Myriokefalitakis et al., (2016) published as discussion paper in Biogeosciences may consider these results from the northern Indian Ocean.

---

## Referee Comment (RC1) · Anonymous Referee #1 · 1 Jul 2016

The atmospheric deposition of P provides an important source of P to the terrestrial/marine ecosystems with notable effects. This manuscript studies the atmospheric cycle of P using a global 3-D chemistry-transport model. In particular, this study accounts for the P mobilization from mineral dust, which is found to be an important source of dissolved P. I feel that this paper can be published after addressing some comments below. Specific comments: 1) Line 23 (p1): BP and DP are confusing in the abstract. 2) Line 25 (p1): It is unclear how the <50%> uncertainty is quantified in the results. 3) Line 26-29 (p1): It is better to give the dissolution fluxes, rather than one percentage. 4) Line 30 (p1): "dissolution flux of P" and "P mobilization flux" are confusing in the abstract. 5) Line 9 (p5): Using a coarse-resolution model to get the horizontal

distributions of P concentration and P deposition can lead to substantial biases in the model-observation comparison, which should be considered. 6) Line 10 (p5): It seems that the model is only run with meteorology for one year (2008). In this case, the question is that there is inconsistency if the observations are derived for other years. It means that the interannual variation of P emissions from mineral dust and sea-spray (related to wind) and the episodic transport of P in the atmosphere (related to wind and wet precipitation) cannot be represented in this study. I expect that the model can be run for more years to get an unbiased estimation of P emissions from mineral dust and sea-spray. 7) Equation 1: I am curious to know how the authors get the fraction of P in the emitted dust. The global soil mineralogy datasets are developed for the clay and silt fractions of soils. Therefore, according to my knowledge, the mineralogy of soil is different from the mineralogy of dust. Please explain how the mineralogy of soil is transferred to the mineralogy of emitted dust in the model. 8) Line 29 (p5): I am curious to know what aerosol scheme is used to treat the size evolution of P-containing particles in this model. Is it following a modal method or a sectional method? I expect that the size (0.34 um or 1.75 um) is not fixed in the model. If the size was fixed, I would have to say that the authors should take a more advanced scheme before the paper can be published. 9) Line 30 (p5): Please explain how the solubility of P (10%) is derived? 10) Line 1-5 (p6): Please list the detailed P/BC mass ratios in the emissions (in the Supplementary Materials). I would expect different ratios for emissions from different types of fuel (e.g. coal, oil and biomass, etc). 11) Line 8-11 (p6): Please explain how the initial size distributions of P emissions are treated in the two estimations 1) based on P/BC ratios and 2) based on the new estimation by Wang et al. 12) Line 14 (p6): What is "TP coarse aerosol emissions"? Please replace this with a more rigorous term. 13) Line 20 (p6): I do not think that this assumption is justified. Carbonaceous aerosol (e.g. BC) is mainly contained in fine particles and thus the ageing via coagulation and condensation is very fast. However, P could be more concentrated in coarse particles (as noted by the authors) and the ageing should be slower. 14) Line 23 (p7): Please compare the global total sea-spray emissions output from TM4-ECPL to other

estimations before using the data. It seems that the super-coarse mode of sea-spray has not been considered in this model, and it might lead to significant underestimation of P concentrations when comparing with cruise measurements over the oceans. 15) Line 30-35 (p7): The authors are right to use the surface concentrations of Na and PO4. I am curious to know how deep is defined as the surface water for Na and PO4. It is better to clarify this, although it is not easy to make sure that they are consistent. 16) Line 14 (p8): I am not against what is done here, but I would like to make it clearer that we should always be very careful when taking this kind of assumption. For example, the authors took the sulphur emissions from Andres and Kasgnoe (1998) while adopting the size distribution proposed by Dentener et al. (2006). Are they consistent? 17) Equation 6: I am curious to know what H+ activity is used in this equation. The authors seem to be clear that the H+ activity in aerosol water is different from that in cloud droplet. Which one is used here?, or both are used. Please clarify it. 18) Section 3.2: Please give maps of the geographic distributions of sites measuring P-containing aerosols concentrations and dry deposition fluxes in the Supplementary Materials to show the spatial coverage of the observational data used in this study. 19) Line 8-13 (p11): Since the model is run with meteorology for only one year (2008), I am curious to know how the authors can compare the modeled P concentrations in 2008 with that measured in other years. If the measurements were also all measured in the year 2008, it should be fine. 20) Line 8 (p11): It is unclear if the authors evaluated the dry deposition fluxes, or they have evaluated both the dry and wet deposition. Please make it consistent. 21) Line 20-23(p11): Here, I am not convinced of the conclusion. Accordingly to our knowledge, the emission inventory is important for the modeling of P, but it is not the only factor that matters. For example, the treatment of aerosol scheme and the initial size distribution of P in the emissions can also influence the concentrations and transport of P. Unfortunately, the authors do not provide necessary information on these in their methods, making it hard to judge whether the conclusion is right or not. As I know from Wang et al., they have accounted for three size bins of P-containing particles in their model, rather than the two size bins in your model. So, it is at least

necessary to repeat the treatment of size distributions by Wang et al. and discuss on the impact. 22) Line 27 (p11): I would expect that the authors compare the modeled P concentrations with that from cruise measurements for the same days (see Figure 8 in Wang, R. et al. Atmos. Chem. Phys., 15, 6247-6270, 2015). 23) Line 27 (p13): "SOx, NOx and NHx anthropogenic" -> " anthropogenic SOx, NOx and NHx". 24) Section 4: This part is interesting and novel. It will be better if the authors can add discussion on what can be improved to get a better understanding of this impact in the future studies or what is the most uncertainty.

―――――――――――――――――――――――

---

## Referee Comment (RC2) · Anonymous Referee #2 · 5 Jul 2016

Review of Myriokefalitakis,

This is a potentially publishable paper looking at an interesting topic. With a few revisions in the presentation, the paper should be publishable.

Major points: 1. This is the first study which explicitly models the evolution of the state of the phosphorus in the atmosphere, which is an important innovation. I think you should highlight this in the abstract, introduction and conclusions. 2. More careful consideration of size. As shown by the contrast between Wang et al., 2014 and [J Brahney et al., 2015], there is a large sensitivity in the budgets of P to assumptions about aerosol size. Please discuss in the methods section the sizes you are considering within the model. Please also discuss the deposition data, and whether it includes

sizes of particles you don't include, and so some part of that mass should be neglected (e.g. Brahney et al., 2015 discussion of sizes). 3. More description of the observations and how you are comparing to them. Please add a section of the methods talking about your observations. It's unclear in your scatter plot where the data comes from and how you are comparing the deposition data (e.g. are the sizes consistent?). Could you show on one of your plots where the data comes from in these scatter plots (a little x?), and maybe show your observations from the cruises in a different color than the observations from the station on the scatter plot? Please discuss a little bit the differences in these observations and their value for your comparison (e.g. temporal variability vs. cruises showing spatial+temporal variability). Also, please include a description of your metric within the methods section (not a reference to another paper in the results section). 4. Compare % soluble observations vs. model? You get really high solubilities far away from the sources—is there any evidence of this? Perhaps if you compare % solubility instead of soluble P amount, it might make your case more compelling that you are doing the solubility right? In a sense the P amounts are dominated by getting the P sources, and the right size comparisons, but the solubility, which is the real innovation in this study, might be better explored by the % solubility in the obs. Vs. model? Even if there is no evidence of these high solubilities, you are underestimating soluble P, so maybe underestimating % solubility close in, so maybe we should believe these high solubilities? 5. You make the case that your results suggest a more important soluble P sources from biogenic aerosols. Why do you get a larger source than previous studies? Is it because you assume more of the biogenic are bioavailable, or are your sources larger? Just add a sentence or two on this.

Details: "primary and secondary sources of P" Are there any secondary sources of P in your approach?

"Okin et al. (2011) evaluated the impact of Fe and P atmospheric deposition to the ocean in increasing N2-fixation and found that Fe deposition is more important than P deposition in supporting N2-fixation, while they pointed out the large uncertainty in the

bioavailability of atmospherically deposited P." There are also ocean biogeochemical model studies which show this same results either: [A Krishnamurthy et al., 2010] or [R Wang et al., 2015] which also suggest that atmospheric deposition of P doesn't matter because of large P reservoirs in ocean.

"Wang et al. (2014) taking into account the potential volatilized-P produced during combustion processes, calculated about 30 times higher global atmospheric P emissions from biomass burning and anthropogenic combustion processes (0.7 Tg-P yr-1 and 1.8 Tg-P yr-1 respectively)." This is not accurate. Tipping et al., 203 put together a compilation of deposition in ecosystems, and indicated that the observations suggest this deposition dominated by locally generated primary biogenic material, in the aerosol mode >10um which is not long range transported. Wang et al., 2014 used the mismatch in size between their <20um modeled aerosols and the observations in the source regions and assumed that this mismatch was only from combustion sources. Thus there is a serious methodological problem in the Wang et al., study, and they don't bother to compare against the available concentration data which would have revealed this problem (as you do here), nor the observation-based source apportionment in Mahowald et al., 2008, which was consistent with the much smaller combustion sources. Instead one should say perhaps: Wang et al. (2014) taking into account the potential volatilized-P produced during combustion processes by assuming that all mismatches between observed deposition (<1000um aerosols) and modeled-long-range transported (<20um) deposition was due to combustion, estimated about 30 times higher global atmospheric P emissions from biomass burning and anthropogeniccombustion processes (0.7 Tg-P yr-1 and 1.8 Tg-P yr-1 respectively)." Or simply don't refer to that paper here or mention it in passing, since it is deeply methodologically flawed. [J Brahney et al., 2015] discusses how to compare to the Tipping et al., data in a more explicit aerosol size manner, and extends the Mahowald et al., 2008 study, showing that one can match deposition and concentration observations at the same time.

"where EDu is the on-line calculated dust emissions in the model, F880 is a factor

applied to adjust the P emissions to the global mean P content of mineral dust in the model domain of 880 ppm per weight as observed by Zamora et al. (2013), and EP is the resulted inorganic P emissions from mineral. P-containing minerals associated with dust particles are emitted in the fine and coarse mode with mass median radii (lognormal standard deviation) of 0.34 $\mu$m (1.59) and 1.75 $\mu$m (2.00), respectively. The apatite emissions from mineral dust calculated for the year 2008 amount to 1.034 Tg-P yr-1 30 with 10% of it (0.103 Tg-P yr-1) in the dissolved form (Table 1)." How does this approach compare to the size resolved methods used in [J Perlwitz et al., 2015] for this mineral?

Section 2.0: model description; can you describe your aerosol size bin or modal structure for the primary aerosols in P?

Section 2.1.3: how do you include bits of insects and plants that would be part of pbap? How important is the neglect of these terms to your budgets do you think?

Please fix English by adding preposition (e.g. of): "(i.e. Nigeria downwind the Sahara Desert, Pakistan downwind the Thar Desert and China downwind of the Gobi desert)."

"In Fig. 4b, PO4 deposition fluxes (wet and dry deposition) from the Vet et al. (2014) compilation and from observations at Finokalia Station (Mihalopoulos and co-workers, unpublished data) are also compared with the model derived fluxes for the PRESENT simulation." What is the size distribution of the PO4 in the deposition? Is it the same size as the modeled boxes? I also think you should present the data you are going to compare against as a section in the methods, and describe the characteristics of the data, especially as some of the data is from unpublished sources. We also need to know where this data comes from physically: is it all in Greece, or elsewhere?

"(MNB; see definitions of statistical parameters in Myriokefalitakis et al. (2015))" You also need to describe your methods in the methods section: it is not ok to refer us for basic information to another paper.

Figure 7: maybe you want to reformat so that there won't be too much white space in the final figure for this?

"The present-day P solubility of deposited aerosols (hereafter SP = %DP/TP) is calculated to vary spatially significantly (Fig. 7a)," vary spatially significantly is awkward: please rephrase and only use significantly if you mean statistically significantly.

For your past and future estimates: Your P is strongly dependent on dust, and yet you don't include any changes in dust. I don't think you need to add much here, but just some statements that dust appears to vary strongly and perhaps be sensitive to humans climate change and/or land use [P Ginoux et al., 2012; N Mahowald et al., 2010; J Prospero and P Lamb, 2003], and thus that would also be an important drivers of changes in P and SP.

Brahney, J., N. Mahowald, A. Ballantyne, and J. Neff (2015), Is atmospheric phosphorus pollution altering global alpine lake stoichiometry? , Global Biogeochemical Cycles, 29, doi:10.1002/2015GB005137. Ginoux, P., J. Prospero, T. E. Gill, N. C. Hsu, and M. Zhao (2012), Global scale attribution of anthropogenic and natural dust sources and their emission rates based on MODIS deep blue aerosol products, Reviews of Geophysics, 50, (RG3005, ), DOI:10.1029/2012RG000388. Krishnamurthy, A., J. K. Moore, N. Mahowald, C. Luo, and C. S. Zender (2010), The Impacts of atmospheric nutrient inputs on marine biogeochemistry, Journal of Geophysical Research, 115(G01006), doi:10.1029/2009JG001115. Mahowald, N., et al. (2010), Observed 20th century desert dust variability: impact on climate and biogeochemistry, Atmospheric Chemistry and Physics, 10(doi:10.5194/acp-10-10875-2010), 10875-10893. Perlwitz, J., C. Perez, and R. Miller (2015), Predicting the mineral compoosition of dust aerosols –Part 1: representing key processes, Atmospheric Chemistry and Physics 15, 11593-11627, doi:11510.15194/acp-11515-11593-12015. Prospero, J., and P. Lamb (2003), African droughts and dust transport to the Caribbean: climate change implications, Science, 302, 1024-1027. Wang, R., et al. (2015), Influence of anthropogenic aerosol deposition on the relationship between oceanic producitivty and warming, Geophysical Research Letters, 42, 10,745-710,754; doi:710.1002/2015GL066753.

---

## Author Comment (AC1) · 31 Oct 2016

We thank the reviewer for the careful reading of the paper. Please find below the point-by-point answers to the reviewer's comments:

**Specific comments:**

**1) Line 23 (p1): BP and DP are confusing in the abstract.**
        We rephrased the sentence in the abstract, by removing the abbreviation BP.

**2) Line 25 (p1): It is unclear how the <50%> uncertainty is quantified in the results.**
        "As explained in the $2^{nd}$ paragraph of section 3.2, based on the statistics of the comparison between the model results and the measurements of TP and PO4 aerosol concentrations the NMB is found to be about -67%.   This sentence has been rephrased to '…compared with available observations, indicating however an uncertainty of about 70% on current knowledge of the sources that drive P atmospheric cycle'.

**3) Line 26-29 (p1): It is better to give the dissolution fluxes, rather than one percentage.**
        We have chosen to provide the percentage change in order to show a quantitative comparison with the present day flux for which the absolute number is given earlier in line 21. The absolute fluxes for PAST and FUTURE simulations can be easily derived from these numbers. Therefore, no change has been made to this sentence.

**4) Line 30 (p1): "dissolution flux of P" and "P mobilization flux" are confusing in the abstract.**
        To avoid confusion we now use only the term 'P solubilisation flux'.

**5) Line 9 (p5): Using a coarse-resolution model to get the horizontal distributions of P concentration and P deposition can lead to substantial biases in the model-observation comparison, which should be considered.**
        We re-run the model in its finer horizontal resolution (i.e. $3^{o}$ x $2^{o}$ in longitude by latitude) and updated the manuscript and the respective figures.

**6) Line 10 (p5): It seems that the model is only run with meteorology for one year (2008). In this case, the question is that there is inconsistency if the observations are derived for other years. It means that the interannual variation of P emissions from mineral dust and sea-spray (related to wind) and the episodic transport of P in the atmosphere (related to wind and wet precipitation) cannot be represented in this study. I expect that the model can be run for more years to get an unbiased estimation of P emissions from mineral dust and sea-spray.**
        Our study has to be seen as a 'climatological' rather than a year specific study. This is justified by the large uncertainty associated with the sources, the dissolution kinetics and the deposition parameterisation. There is indeed a year-to-year variability in the emissions and atmospheric transport that could be addressed by performing multiple year simulations. However, such simulations are computational intensive and the added value to the main objective of the paper is very small. Furthermore, a model-to-observations comparison for a specific year limits the number of available measurements and thus the statistical significance of the comparison. However, to satisfy the reviewer's request, we run our model for a longer period that covers a significant number of available measurements to make year specific comparisons. This has been the main reason for the delay in the revision of our manuscript.

**7) Equation 1: I am curious to know how the authors get the fraction of P in the emitted dust. The global soil mineralogy datasets are developed for the clay and silt fractions of soils. Therefore, according to my knowledge, the mineralogy of soil is different from the mineralogy of dust. Please explain how the mineralogy of soil is transferred to the mineralogy of emitted dust in the model.**
        For the present study, we applied the available P- content distribution of soils on dust emissions making no distinctions between soil and dust particles. We estimated the P-content in dust based on the database by Nickovic et al. (2012) that only provides the mass fraction of P in soils. The emissions of dust were calculated as a function of wind velocities at a height of 10 m over dust source locations and soil types (Tegen et al., 2002) for two modes accumulation and coarse (van Noije et al., 2014) The resulted P emissions were scaled to a global mean of 880 ppm per weight as observed by (Zamora et al., 2013). This

is explained in section 2.1.1. Thus, as in most relevant studies but one (Perlwitz et al. (2015)), air borne dust particle emissions are assumed to have the same mineralogy with that of the soil from which they originate.
We have further added the relevant following discussion in section 2.1.1.

"Although in most relevant modelling studies airborne P-containing dust particle emissions are assumed to have an average P content of 720 ppm (Mahowald et al., 2008; Wang et al., 2014; Brahney et al., 2015), in the atmosphere due to transport, ageing and deposition processes the overall mineralogy may change the chemical composition and size of dust aerosol population. In a recent iron modelling study however (Perlwitz et al., 2015), a significant effort has been made to model the mineral composition of dust considering the differences from the original soil composition. Perlwitz et al. (2015) have found significant overestimate (a factor of 10-30) mainly in the fine aerosol emissions that are the smallest part of dust emissions (e.g. about 7% of the total emissions in our model) and an underestimate in the larger particles emissions both for total dust and for individual minerals when the mineralogy of dust aerosol is assumed to be the same as that of the soil. However for the present study, we did not account for different P content for dust particles in the fine and the coarse mode, since the global soil mineralogy dataset used (Nickovic et al., 2012) does not provide any information of P content in silt and clay soil particles separately. Note also, that recent studies indicate that dust super-coarse particles can be very important for the biogeochemistry over land, since they can represent the dominant fraction of dust close to source regions (Lawrence and Neff, 2009) (Neff et al., 2013). (Brahney et al., 2015) modelling study that focused on the atmospheric phosphorus deposition over global alpine lakes, based on Neff et al. (2013) observations, estimated that only 10% of the mass that travels in the atmosphere is within the <10 μm size fraction. In our study we do not account for super-coarse dust particles because due to their short atmospheric lifetime, they are emitted and deposited in the same model grid box (Brahney et al., 2015). This omission is not expected to have significant impact on our results, since the present work is focused on the P-solubilisation mechanisms occurring via atmospheric long-transport mixing and on the bioavailable P deposition over the marine environment. "

**8) Line 29 (p5): I am curious to know what aerosol scheme is used to treat the size evolution of P containing particles in this model. Is it following a modal method or a sectional method? I expect that the size (0.34 um or 1.75 um) is not fixed in the model. If the size was fixed, I would have to say that the authors should take a more advanced scheme before the paper can be published.**
The model follows the modal approach to represent aerosol sizes. Specifically, the model configuration used for this study is focused on the chemical interactions of dust particles and the acid P-solubilisation of apatite minerals and, thus, on chemistry and on aerosol mass simulations. The strong point of our model is exactly the representation of atmospheric chemistry in all phases, gas, aqueous and aerosol phase. **Overall 32 model P-containing aerosol tracers are used to represent phosphorus** in the model of different sizes and solubilities. In TM4-ECPL, different sources emit P-containing aerosols of different sizes represented by lognormal distributions as summarized in the Table below.

Table : P-sources and aerosol size lognormal modes taken into account in TM4-ECPL model.

| P-sources | Lognormal modes - Dry number median radii in um (sigma) | |
|---|---|---|
| Dust | 0.34* (1.59) | 1.75* (2.0) |
| Combustion | 0.04 (1.8) | 0.50 (2.0) |
| Primary biological aerosol particles | | 1.50 (monodisperse) |
| Volcanoes | 0.04 (1.59) | |
| Sea-spray | 0.09 (1.59) | 0.794(2.0) |

(*) Mass mean radii

For each aerosol mode and source (Figure 1 of the manuscript) the model accounts for total P, phosphate, insoluble and soluble OP, for the dust source it also accounts for the two P-containing minerals (fluoroapatite and hydroxyapatite) as described in the section 2.1.1. These are individually transported, aged and deposited in the atmosphere. The 'dry' aerosol hygroscopic growth in the model is treated as a function of ambient relative humidity and the composition of soluble aerosol components based on experimental work by Gerber (1985; 1988) and this uptake of water on aerosols changes the particle size. In addition, during atmospheric transport there are major changes in the size distribution of aerosols as a

consequence of the removal of larger particles due to gravitational settling. The P-containing aerosols follow the same parameterizations, hygroscopic growth and removal processes are assumed to affect the mass median radius (i.e. size).

Note also that although our model does not include the most sophisticated aerosol scheme (uses the modal lognormal distribution instead of the more compute demanding sectional approach), TM4-ECPL is the first CTM model that takes into account the P-solubilisation due to atmospheric acidity, instead of taking a constant solubility fraction. In addition, TM4-ECPL accounts for all known sources of atmospheric phosphorus including the primary biological particles. These strongly innovative aspects of our study deserve publication.

Relevant text has been added in section 2 and before section 2.1. "To represent phosphorus in the model, overall 32 model P-containing aerosol tracers are used of different sizes and solubilities. In TM4-ECPL, different sources emit P-containing aerosols of different sizes represented by lognormal distributions as outlined in section 2.1. For each aerosol mode and source (Figure 1 of the manuscript) the model accounts for total P, phosphate, insoluble and soluble OP. For the dust source it also accounts for the two P-containing minerals (fluoroapatite and hydroxyapatite) as described in the section 2.1.1. These are individually transported, aged and deposited in the atmosphere. The 'dry' aerosol hygroscopic growth in the model is treated as a function of ambient relative humidity and the composition of soluble aerosol components based on experimental work by Gerber (1985; 1988) and this uptake of water on aerosols changes the particle size. In addition, during atmospheric transport there are major changes in the size distribution of aerosols as a consequence of the removal of larger particles due to gravitational settling. The P-containing aerosols follow the same parameterizations, hygroscopic growth and removal processes are assumed to affect the mass median radius (i.e. size)."

**9) Line 30 (p5): Please explain how the solubility of P (10%) is derived?**

The soluble fraction used in our model is based on the measurements of leachable inorganic phosphorus (LIP) for Saharan soil dust, as presented by Nenes et al. (2011). These authors found that LIP represented up to 10 % of total inorganic P in Saharan soil samples and dry fallout collected during Sahara dust storms before acid-treatment. Moreover, Yang et al. (2013) estimated the labile inorganic P in the top soil on the global scale at about 3.6 Pg-P that corresponds to about 10% of the estimates of total soil P on the global scale 30.6-40.6 Pg-P (Smil, 2000; Wang et al., 2010; Yang et al., 2013). To further investigate uncertainties associated with the soluble fraction of P-containing dust aerosol emissions in our model, an additional simulation has been performed neglecting any soluble fraction on initial emissions.

The above discussion has been added in section 2.1.1.

The comparison of model results to the observations shows no significant change in the performance of the model due to this additional soluble P primary source. On the contrary, our results presented in Figure S4 (of the discussion paper) show significant impact of the secondary P-solubilisation source, increasing the soluble P simulated concentrations. Relevant discussion has been added in the revised version of the manuscript in section 3.2.
"Neglecting the P dissolution definitely degrades the comparisons of model results with observations. On the other hand the results show very small sensitivity to the assumption of soluble fraction of the primary emissions of P. This finding supports the importance of the atmospheric processing of dust for the atmospheric DP cycle as well as the potential underestimate of the DP source in all sensitivity simulations. Such underestimate could be associated with an underestimate in the primary source or in the secondary (atmospheric processing) of DP and deserves further studies. "

**10) Line 1-5 (p6): Please list the detailed P/BC mass ratios in the emissions (in the Supplementary Materials). I would expect different ratios for emissions from different types of fuel (e.g. coal, oil and biomass, etc).**
        As clearly stated in section 2.1.2 lines 2 and 3, for the present work, we used the P/BC combustion emissions factors of Mahowald et al. (2008) that are based on BC aerosol emissions for the fine and coarse mode (i.e. P/BC in fine mode of 0.0029 and P/BC in coarse mode of 0.02; independent of fuel type

(Mahowald et al., 2008, 2005). However, we know that recently new developments on P combustion emissions are published by Wang et al. (2014) following a more comprehensive approach, accounting for different emission factors per fuel type. That estimate is significantly higher than what is derived based on P/BC emission factors and BC emissions. Therefore, as stated at the end of the first paragraph of section 2.1.2, we have performed an additional present-day sensitivity simulation using the P-combustion emissions developed by Wang et al. (2014) (after personal communication with the author). The results of this simulation are discussed in section 3.2 and plotted in the supplementary figures S4-S6.

We comment on the sensitivity simulations in the manuscript as follows (section 3.2):
"In Figures S4 and S5 (supplement) are also presented the results of sensitivity simulations and the base case simulation with the aerosol observations and dry deposition fluxes, respectively, while figures S6 also show the comparison of the annual cycles of the atmospheric concentrations (TP and PO4) and deposition fluxes (dry and wet deposition), against the TM4-ECPL monthly model results. For cruise measurements over the Atlantic and Pacific Oceans (Baker and Croot, 2010; Martino et al., 2014; Powell et al., 2015), and the global compilation of deposition rates (Vet et al., 2014), the observations are also spatially averaged inside the same model grid box. These comparisons show almost similar performance for all sensitivity simulations but one falling in most cases close to the lower edge of observed concentrations and deposition fluxes. However, taking into account the Wang et al. (2014) P-combustion sources, the model performs better over the land (e.g. for TP concentrations at Corsica; Fig. S4g, and for DP concentrations at the Finokalia monitoring station; Fig. 6b,f,i), indicating that the base simulation underestimates either anthropogenic combustion sources or other natural P sources. Neglecting P dissolution definitely degrades the comparisons of model results with observations. On the other hand the results show very small sensitivity to the assumption of soluble fraction of the primary emissions of P. This finding supports the importance of the atmospheric processing of dust for the atmospheric DP cycle as well as the potential underestimate of the DP source in all sensitivity simulations. Such underestimate could be associated with an underestimate in the primary source or in the secondary (atmospheric processing) of DP and deserves further studies."

Note however that the present study focuses on the natural emissions of phosphorus and how these are affected by human emitted acidic substances. This study does not fully cover all aspects of phosphorus cycle since many questions remain open for future work.

**11) Line 8-11 (p6): Please explain how the initial size distributions of P emissions are treated in the two estimations 1) based on P/BC ratios and 2) based on the new estimation by Wang et al.**
  See further our reply to point 21

**12) Line 14 (p6): What is "TP coarse aerosol emissions"? Please replace this with a more rigorous term.**
  This sentence has been rephrased: 'BC emissions from anthropogenic combustion in the coarse mode are assumed to be 25% of those in the fine mode (Jacobson and Streets, 2009), while biomass burning emissions in the coarse mode are assumed equal to 20% of those of fine aerosols (Mahowald et al., 2008).'

**13) Line 20 (p6): I do not think that this assumption is justified. Carbonaceous aerosol (e.g. BC) is mainly contained in fine particles and thus the ageing via coagulation and condensation is very fast. However, P could be more concentrated in coarse particles (as noted by the authors) and the ageing should be slower.**
  We thank the reviewer for pointing us this misleading sentence. In our model simulations, we assumed that the insoluble fraction of organic phosphorus (OP) associated with primary emissions of organic aerosol is converted to dissolved OP (DOP) during atmospheric ageing, based on the (Tsigaridis and Kanakidou, 2003) parameterizations for carbonaceous aerosols that were using a mean mass aerosol size of 0.25 µm and a monolayer thickness of 2.5 nm of the particle surface. For the present study, we take into account the different aerosol sizes (based on the lognormal distributions) of hydrophobic OP aerosols

to compute their conversion rates to hydrophilic aerosols. Note that based on that parameterisation the conversion rate is reduced with increasing particle size.

To avoid further misunderstandings, we rephrase this sentence as "The insoluble fraction of OP associated with combustion emissions can be further converted to soluble OP (DOP) during atmospheric ageing, using the ageing parameterization for primary hydrophobic organic aerosols in the model (Tsigaridis and Kanakidou, 2003; Tsigaridis et al., 2006), but for the respective size and lognormal distribution of OP aerosols with the larger particles experiencing the smallest conversion rates"

**14) Line 23 (p7): Please compare the global total sea-spray emissions output from TM4-ECPL to other estimations before using the data. It seems that the super-coarse mode of sea-spray has not been considered in this model, and it might lead to significant underestimation of P concentrations when comparing with cruise measurements over the oceans.**

Sea-spay emissions are computed by the model using the parameterization developed and tested by Vignati et al. (2010) and also used in Myriokefalitakis et al. (2010). As in most models, these emissions are distributed into fine and coarse modes, super coarse sea-salt particles are not considered. Emissions are driven by the model's meteorology. For the year 2008 (base year for this study), TM4-ECPL model calculates a total of 7278 Tg $y^{-1}$ of sea-salt emissions from which 36 Tg $yr^{-1}$ are fine mode particles. These numbers compare well (by 10% lower) with the AEROCOM recommendation of 7925 Tg $yr^{-1}$ by Dentener et al. (2006) and are within the range of 2272-12462 Tg $y^{-1}$ computed by (Tsigaridis et al., 2013) using different parameterisations in the GISS modelE.

Relevant discussion is now added in section 2.1.4 right after the first sentence.

"Sea-spray emissions are driven by the model's meteorology and for the year 2008 the model calculates a total of about 8284 Tg $yr^{-1}$ of sea-salt emissions (of which 41 Tg $yr^{-1}$ are in the fine mode). These numbers compare well with the AEROCOM recommendation of 7925 Tg $yr^{-1}$ by Dentener et al. (2006) and are within the range of 2272-12462 Tg $yr^{-1}$ computed by Tsigaridis et al. (2013) using several different parameterisations. Note that our sea-salt source estimation is however much lower than the one used in the modelling study by Wang et al. (2014) (i.e. 25300 Tg $yr^{-1}$), since super coarse sea-salt particles are not considered in the current parameterization."

However, this omission can explain some discrepancies between model results and observations only when these later concern bulk aerosols in oceanic regions, so they could include super-coarse particles, which is the case for wet or dry deposition samples. In many cases, aerosol samples have been collected with inlet devices that enable collection of specific fractions of aerosols that cut super-coarse particles. When bulk aerosols have been collected, then the presence of super-coarse aerosols might introduce discrepancies between model results and observations. To distinguish such differences, we separated the bulk aerosol observations from the size segregated ones in section 3.2 and in Figures S4, S6.

And at the end of the section 2.1.4 we have also added the following comment: "The omission of the super coarse sea salt aerosol might affect our estimates of P deposition to the ocean. Brahney et al (2015) evaluated this source at 0.0046 Tg-P $yr^{-1}$, an amount that introduces a 3% underestimate to the here calculated present-day P deposition flux to the oceans."

**15) Line 30-35 (p7): The authors are right to use the surface concentrations of Na and PO4. I am curious to know how deep is defined as the surface water for Na and PO4. It is better to clarify this, although it is not easy to make sure that they are consistent.**

We use the consistent dataset for surface concentrations of Na and PO4 from the LEVITUS94 Ocean Climatology database that we have downloaded from the webpage that is provided in the manuscript and is active (e.g. http://iridl.ldeo.columbia.edu/SOURCES/.LEVITUS94/.ANNUAL/.PO4/). Specifically the surface concentrations correspond to the data labelled as for 0m depth, while the next depth with available data is 10m. The same depth has been chosen for the salinity distribution that was used to derive Na concentrations. We now specify this in the manuscript (section 2.1.4).

**16) Line 14 (p8): I am not against what is done here, but I would like to make it clearer that we should always be very careful when taking this kind of assumption. For example, the authors took the sulphur emissions from Andres and Kasgnoe (1998) while adopting the size distribution proposed by Dentener et al. (2006). Are they consistent?**

(Dentener et al., 2006) emissions for the AEROCOM project are also based on the GEIA inventory for sulphur emissions by (Andres and Kasgnoc, 1998).

**17) Equation 6: I am curious to know what H+ activity is used in this equation. The authors seem to be clear that the H+ activity in aerosol water is different from that in cloud droplet. Which one is used here? or both are used. Please clarify it.**

P-dissolution is calculated for aerosol water and cloud droplets separately. In the model we separate these two pathways due to the different properties of the solution. Since the aerosol solution is more condensed than the cloud droplets, we use the term 'H$^+$ activity' in contrast to cloud droplets where the H$^+$ activity can be considered equal to the concentrations. For aerosol water, the activity of H$^+$ is calculated on-line in the model by the thermodynamic module ISORROPIA II (Fountoukis and Nenes, 2007). For cloud water, H$^+$ concentration is calculated by the aqueous-phase chemistry module as presented in Myriokefalitakis et al. (2011; 2015). For clarity we have added the last two sentences in the second paragraph of section 2.2.

**18) Section3.2: Please give maps of the geographic distributions of sites measuring P-containing aerosols concentrations and dry deposition fluxes in the Supplementary Materials to show the spatial coverage of the observational data used in this study.**

New figures 4a and 4b have been added to provide the location of the observations.

**19) Line 8-13(p11): Since the model is run with meteorology for only one year (2008), I am curious to know how the authors can compare the modeled P concentrations in 2008 with that measured in other years. If the measurements were also all measured in the year 2008, it should be fine.**

Please see answer to reviewer's comment # 6.

**20) Line 8 (p11): It is unclear if the authors evaluated the dry deposition fluxes, or they have evaluated both the dry and wet deposition. Please make it consistent.**

We evaluated separately the dry and wet deposition fluxes, as explained in section 3.2. To limit the number of figures we have used one figure (Figure 4) to show both comparisons but we used different colours to distinguish the dry deposition fluxes (in red) from the wet deposition fluxes (in green). However for clarity we have removed the wet deposition fluxes comparison from this figure and we now present such comparison in the supplement: Figures S5c,d and Figures S6f, h.

**21) Line 20-23(p11): Here, I am not convinced of the conclusion. Accordingly to our knowledge, the emission inventory is important for the modeling of P, but it is not the only factor that matters. For example, the treatment of aerosol scheme and the initial size distribution of P in the emissions can also influence the concentrations and transport of P. Unfortunately the authors do not provide necessary information on these in their methods, making it hard to judge whether the conclusion is right or not. As I know from Wang et al., they have accounted for three size bins of P-containing particles in their model, rather than the two size bins in your model. So, it is at least necessary to repeat the treatment of size distributions by Wang et al. and discuss on the impact.**

We thank the reviewer for this remark. In TM4-ECPL model for all our simulations we consider two sizes of combustion aerosols, fine and coarse modes, assuming lognormal distributions for each mode (see also our reply to reviewer's comment #8). This is described in the beginning of the second paragraph of section 2.1.2 of the discussion paper, i.e.: 'In the model, a number mode radius of 0.04 μm and a lognormal standard deviation of 1.8 are assumed for fine P emissions, while for coarse P a number mode radius of 0.5 μm and lognormal standard deviation of 2.00 are used as proposed for combustion aerosols by Dentener et al. (2006)'.

For the sensitivity study with the database of the P combustion emissions recently developed by Wang et al. ( 2014), we now consider 3 modes, following the method described in the supplement of that publication. This is now done and clarified at the end of section 2.1.2.

"To further investigate uncertainties in the P combustion emissions in our model, an additional present-day simulation was performed taking into account the total (bulk) mass of anthropogenic combustion and biomass burning P emissions, as developed by Wang et al. (2014) (R. Wang, personal communication, 2016). According to that database, global anthropogenic emissions from fossil fuels, biofuels and

deforestation fires amount to 1.079 Tg-P yr$^{-1}$ and natural fire emissions equal to 0.808 Tg-P yr$^{-1}$. For this sensitivity simulation, we apply the size distribution as described in Wang et al. (2014); i.e. by dividing total emissions into 3 modes - one fine (2% of P) and two coarse modes (25% and 73% of P) - with mass mode dry diameters of 0.14 μm, 2.5 μm and 10 μm and lognormal standard deviations of 1.59 and 2.00 for fine and coarse modes, respectively."

**22) Line 27 (p11): I would expect that the authors compare the modeled P concentrations with that from cruise measurements for the same days (see Figure 8 in Wang, R. et al. Atmos. Chem. Phys., 15, 6247-6270, 2015).**
> In the discussion paper we compared simulated and measured concentrations for the same days but not always for the same year since the simulations were done for the year 2008. We now have performed additional 11-year simulations to compare data and model for the specific year and satisfy the reviewer on this point. See also our reply to the comment #6.

**23) Line 27 (p13): "SOx, NOx and NHx anthropogenic" -> " anthropogenic SOx, NOx and NHx".**
> Done.

**24) Section 4: This part is interesting and novel. It will be better if the authors can add discussion on what can be improved to get a better understanding of this impact in the future studies or what is the most uncertainty.**
> We further develop this part in the revised version and in particular we have added some recommendations in section 4.2: "However, large uncertainties are associated with this innovative finding, since the estimates of the global source of PBAPs vary by more than an order of magnitude, their size distributions, their mass density are uncertain and the P-content in these aerosols is also highly variable, spanning 2 orders of magnitude (e.g. (Kanakidou et al., 2012) supplementary material and references therein). All these parameters have to be studied by targeted experiments to improve knowledge of their contribution to the atmospheric P cycle." and further in the discussion "Our results also show that the P solubilisation from dust aerosol during atmospheric transport and mixing with acidic pollutants is important for DP deposition and deserves further kinetic studies to improve parameterisation of the solubilisation kinetics of various P containing minerals as a function of acidity and temperature."

[revised manuscript text omitted]

---

## Author Comment (AC2) · 31 Oct 2016

We thank the referee for the careful reading of the manuscript and the fruitful comments that helped improving the presentation of our study. We have addressed all of them as described in the following point-by point replies to the referee's comments.

**Major points:**

**1)    This is the first study which explicitly models the evolution of the state of the phosphorus in the atmosphere, which is an important innovation. I think you should highlight this in the abstract, introduction and conclusions.**

We thank the reviewer for recognizing the value of our study. The third and fourth sentences of the abstract now read: "The P solubilisation from mineral dust under acidic atmospheric conditions is also parameterized in the model and is calculated to contribute about one third (0.14 Tg-P yr$^{-1}$) of the global DP atmospheric source. To our knowledge, this is the first global study that explicitly models the evolution of phosphorus speciation in the atmosphere."

Furthermore a sentence has been added in the last paragraph of the introduction "To our knowledge, this is the first study that accounts for both inorganic and organic forms of P and their evolution in the atmosphere".

The first sentence in the conclusion has been accordingly modified as follows: "Primary TP and DP emissions accounting for both inorganic and organic P and for the atmospheric processing of P are taken into account for the first time in the state-of-the-art atmospheric chemistry transport global model TM4-ECPL".

**2)    More careful consideration of size. As shown by the contrast between Wang et al., 2014 and [J Brahney et al., 2015], there is a large sensitivity in the budgets of P to assumptions about aerosol size. Please discuss in the methods section the sizes you are considering within the model. Please also discuss the deposition data, and whether it includes sizes of particles you don't include, and so some part of that mass should be neglected (e.g. Brahney et al., 2015 discussion of sizes).**

In our reply to reviewer #1 comment #8, we provide details on the size distribution we consider in the model for the P related tracers. Note we use additional model tracers to represent phosphorus in our model and modal approach to account for the size of the P-containing particles in fine and coarse modes with the mean mass diameters to be dependent on the source categories. Details are provided in our reply to reviewer #1 (comment 8). During atmospheric transport there are major changes in the size distribution of dust as a consequence of the stronger removal of larger particles due to gravitational settling. All details about the modal sizes used in the model to represent P-aerosols are already in the methods section in the source respective P-emissions sub-sections. However a short summary on the treatment of the P-aerosols in our model has been added at the end of the introduction of section 2 on methods. There, we also clarify the number of additional model tracers used to represent P in our model (32) as well as the fact that the size distribution of P-containing aerosols is changing in the model as a result of emissions, atmospheric transport and removal processes.

Furthermore, deposition data in particular wet or bulk dry deposition data include all sizes of aerosols, not only fine and coarse used in TM4 but also super coarse that are deposited close-by their sources, practically in the same grid box of the model where they are emitted.

In the introduction we have modified the 7[th] paragraph of the discussion on the P emissions to provide notions on the size distribution assumption in each source estimate: "the estimates of global strength of the primary P combustion source vary by about an order of magnitude on the global scale, due to the consideration of different forms of the emitted P (i.e. residual or P-containing ash, gaseous or particulate P produced during combustion processes; Wang et al. (2014)) and different size distributions in the emitted P-containing particulate matter. Mahowald et al. (2008) using observed mass ratios of P to Black Carbon (BC) for fine (<2 μm) and coarse (2 μm ≤ mean particle diameter < 10μm) particles, calculated emission fluxes from biomass burning and anthropogenic fuel (i.e. fossil fuel and biofuel) combustion of 0.03 Tg-P yr$^{-1}$ and 0.05 Tg-P yr$^{-1}$, respectively. Tipping et al. (2014) estimated a global atmospheric P emission flux of 3.7 Tg-P yr$^{-1}$ by combining observed deposition rates over land together with modelled deposition rates over the ocean. This emission flux, also accounts for P deposition fluxes of larger particles (e.g. primary biological material, hereafter PBAP, in the aerosol mode >> 10μm) that are mainly deposited very close to their source region and thus not long-range transported. On the other hand Wang et al. (2014), by assuming that combustion processes emit significant amounts of P as large particles > 10μm (hereafter as super-coarse particles) calculated that P

emissions from biomass burning and anthropogenic combustion processes can contribute about 0.7 Tg-P yr$^{-1}$ and 1.8 Tg-P yr$^{-1}$ respectively. In contrast to that study, which was more focused on the impact of anthropogenic combustion on the global P source, Brahney et al. (2015) extended the methodology of Mahowald et al. (2008) in a more explicit aerosol size manner by taking into account also the naturally emitted super-coarse P-containing particles (i.e. dust, PBAP and sea-salt). Brahney et al. (2015) showed that considering this super-coarse fraction as an additional P source, the estimated deposition fluxes close to the source areas where large particles are emitted (e.g. Tipping et al., 2014) can be significantly improved."

Furthermore, in section 2.1.1 we have added the following discussion:
"Note also, that recent studies indicate that dust super-coarse particles can be very important for the biogeochemistry over land, since they can represent the dominant fraction of dust close to source regions (Lawrence and Neff, 2009; Neff et al., 2013). Brahney et al. (2015) modelling study that focused on the atmospheric phosphorus deposition over global alpine Lake, based on Neff et al. (2013) observations, estimated that only 10% of the mass that travels in the atmosphere is within the <10 μm size fraction. In our study we do not account for super-coarse dust particles because due to their short atmospheric lifetime, they are emitted and deposited in the same model grid box (Brahney et al., 2015). This omission is not expected to have significant impact on our results, since the present work is focused on the P-solubilisation mechanisms occurring via atmospheric long-transport mixing and on the bioavailable P deposition over the marine environment."

Therefore the size distribution of the emissions is very important for the model evaluation. Section 2.3 on observations used for model evaluation has been improved to provide information on the sizes of observed aerosols. Similarly such information is provided in section 3.2 on model evaluation. The discussion in this section has been modified to present model evaluation distinguishing for aerosol sizes when available. Figures S4 and S6 have been modified to present size segregated comparisons.
In the present study we do not account for super-coarse dust or sea salt aerosol, while we consider the emissions of pollen that are super-coarse aerosols. Therefore, it is expected that deposition fluxes close to dust source areas are underestimated by the model. Due to the small contribution of sea salt to the P-aerosol budget, the omission of sea salt super coarse aerosol can affect local comparisons but overall does not introduce more than a 3% underestimate of DP flux over the ocean (relevant comment has been added in section 2.1.4 and 3.2).

At the end of section 2.1.4:" The omission of the super coarse sea salt aerosol might affect our estimates of P deposition to the ocean. Brahney et al (2015) evaluated this source at 0.0046 Tg-P yr-1, an amount that introduces a 3% underestimate to the here calculated present-day P deposition flux to the oceans.

In the 3$^{rd}$ paragraph of section 3.2: "The omission of super-coarse marine DP sources associated with sea-salt particles can explain some discrepancies between model results and observations only when these later concern bulk aerosols in oceanic regions (so they could include super-coarse particles), which is the case for wet or dry deposition samples. As discussed in Sect. 2.1.4, this omission can affect local comparisons but overall does not introduce more than a 3% underestimate of DP flux over the ocean. In many cases, aerosol samples have been collected with inlet devices that enable collection of specific fractions of aerosols and eliminate super-coarse particles. When bulk aerosols have been collected, then the presence of super-coarse aerosols might introduce discrepancies between model results and observations. Overall the model performs better for DP dry deposition fluxes over the oceans than over land, indicating a possible underestimate in the continental source of P. "

**3)      More description of the observations and how you are comparing to them. Please add a section of the methods talking about your observations. It's unclear in your scatter plot where the data comes from and how you are comparing the deposition data (e.g. are the sizes consistent?). Could you show on one of your plots where the data comes from in these scatter plots (a little x?), and maybe show your observations from the cruises in a different color than the observations from the station on the scatter plot? Please discuss a little bit the differences in these observations and their value for your comparison (e.g. temporal variability vs. cruises showing spatial+temporal variability). Also, please include a description of your metric within the methods section (not a reference to another paper in the results section).**

Following reviewer's 1 comment 18, we have added two maps in Figure 4a,b that show the location of the measurements. Different measurements (from cruises, stations) have been marked in different colors in the scatter plots. A subsection 2.3 on Data for model evaluation has been added in section 2, where the description of the normalized mean bias (NMB) used to compare model results with observations is now provided. In addition Tables S1 and S2 have been added in the supplementary material to provide information on the species, size, date, location and reference of the observations used for the model evaluation. As we have now performed a simulation for an 11-years period (2000-2010) that covers most of the observational data, the observations are compared to model results that correspond to the day of the observations. In addition, "observations are also spatially averaged inside the same model grid box" (as is mentioned in section 3.2.)

**4)      Compare % soluble observations vs. model? You get really high solubilities far away from the sources. Is there any evidence of this? Perhaps if you compare % solubility instead of soluble P amount, it might make your case more compelling that you are doing the  solubility right? In a sense the P amounts are dominated by getting the P sources, and the right size comparisons, but the solubility, which is the real innovation in this study, might be better explored by the % solubility in the obs. Vs. model? Even if there is no evidence of these high solubilities, you are underestimating soluble P, so maybe underestimating % solubility close in, so maybe we should believe these high solubilities?**

Unfortunately to our knowledge only few observations exist with simultaneous measurements of soluble and total P that provide information on the solubility of P that could be used for such evaluation and most of these are not open ocean data. In addition, the P solubility shown in Figure 7a is computed as the fraction of the sum of the soluble organic and inorganic P to the total P.

The following discussion has been added after the first sentence in section 3.5 on P solubility:
"Over such remote oceanic regions, high solubility fractions are calculated due to low P-containing aerosol mass concentrations, that occur via the long-range transport of fine particles from distance source regions, and the P which is associated with more aged aerosols and thus a greater fraction is present in the soluble mode; either as DIP via mineral acid solubilisation processes or DOP via atmospheric oxidation of P-containing organic aerosols and as PBAPs. Vet et al. (2014) in their review paper for nutrients deposition, also mentioned that the P solubility fractions of wet-only samples on coastal and inland sites have been measured to range from 30% to 90%, reflecting thus the effects of combustion, biomass burning, and phosphate fertilizers on airborne phosphorus concentrations. Anderson et al. (2010) reported that only 15-30 % of P in atmospheric aerosols at the Gulf of Aqaba was water soluble phases or relatively soluble to be bioavailable to the ecosystems. In the Mediterranean the measured median solubilities of the inorganic fraction of P in aerosols (ratio of $PO_4^{-3}$ to total inorganic P) range between 20% and 45% in the East Mediterranean with the lowest values in dust influenced air masses and the highest values in air masses from the European continent (Markaki et al., 2003; Herut et al., 1999) and have been reported to be around 38% in the West Mediterranean (Markaki et al., 2010). However, simultaneous observations of TP and DP deposition fluxes are required to evaluate the solubility fraction of P (both organic and inorganic) over remote oceans and thus to understand the atmospheric fate of P. There are only a few aerosol data available in the literature for the marine atmosphere (Graham and Duce, 1982; Baker et al., 2006a; Baker et al., 2006b; Zamora et al., 2013) that provide hints on the total P solubility. These data indicate P solubilities ranging overall between 0.01% and 94%, with the lowest values corresponding to dust influenced air masses and the highest to sea-salt influenced air masses. Over the northern hemisphere Atlantic ocean P solubilities in aged Saharan dust aerosols have been measured to range from 0.01 to 37% during oceanographic cruises (Baker et al., 2006a;Baker et al., 2006b). At Barbados island median solubilities of P on dust of about 19% and of sea-salt aerosol of about 94% have been reported (Zamora et al., 2013). In the southern Atlantic atmosphere P-solubilities in aerosols of up to 67% (median 8% for dust aerosol and 17% for southern Atlantic aerosol; Baker et al., 2006a) and of up to 87% (median 32%; Baker et al., 2006b) have been reported. These studies but one report P solubility as the ratio of $PO_4^{-3}$ to TP, thus neglecting the organic fraction which has been measured to be about 28-44% (Zamora et al., 2013). Although these observations support high P solubilities in aged aerosols or aerosols impacted by non- dust sources supporting the findings of our modelling study, only the work by (Zamora et al., 2013) could be compared to the here simulated total P solubility (Fig. 7a). They indicate that the model simulated total P solubility is at the upper edge of observed P solubilities.

**5)**      **You make the case that your results suggest a more important soluble P sources from biogenic aerosols. Why do you get a larger source than previous studies? Is it because you assume more of the biogenic are bioavailable, or are your sources larger? Just add a sentence or two on this.**

There was a previous explicit global estimate of this source. The Kanakidou et al. (2012) estimate of OP from PBAPs is of the same order of magnitude with the present estimate. However, that study did not compare with the DIP deposition. In addition Mahowald et al. (2008) estimated total PBAP emissions at 0.164 Tg-P yr$^{-1}$ and considered this amount to be by 50% soluble P (0.082 Tg-P yr$^{-1}$ DIP), while the dust soluble P estimate was 0.115 Tg-P yr$^{-1}$ i.e. of the same order of magnitude with the total PBAPs emissions. So the results are very similar, in our study we are just focusing on the importance of this finding that however needs to be consolidated with additional new observations because both the PBAPs sources and the dust-P solubilisation kinetics are uncertain. In addition here we consider that all biological material is bioavailable.

**Details:**

**1)**      **"primary and secondary sources of P" Are there any secondary sources of P in your approach?**

With 'primary sources' we meant the P (either TP or DP) emissions while with 'secondary sources' we meant the DP released in the atmosphere due to solubilisation processes. To avoid, however, misunderstanding we changed this as remove this part of the sentence and replace it simply by '*P sources*'.

**2)**      **"Okin et al. (2011) evaluated the impact of Fe and P atmospheric deposition to the ocean in increasing N2-fixation and found that Fe deposition is more important than P deposition in supporting N2-fixation, while they pointed out the large uncertainty in the bioavailability of atmospherically deposited P." There are also ocean biogeochemical model studies which show this same results either: [A Krishnamurthy et al., 2010] or [R Wang et al., 2015] which also suggest that atmospheric deposition of P doesn't matter because of large P reservoirs in ocean.**

It is true that the deep ocean is a major source of P for the surface seawater. However depending on season, the water stratification can minimize the impact of the deep water to the upper layers, This is mainly occurring in summer and it is during that period that the atmospheric deposition of P is expected to have the largest impact on the marine ecosystems.

[revised manuscript text omitted]

**4)        Tipping et al., 2003 put together a compilation of deposition in ecosystems, and indicated that the observations suggest this deposition dominated by locally generated primary biogenic material, in the aerosol mode >10um which is not long range transported. Wang et al., 2014 used the mismatch in size between their <20um modeled aerosols and the observations in the source regions and assumed that this mismatch was only from combustion sources. Thus there is a serious methodological problem in the Wang et al., study, and they don't bother to compare against the available concentration data which would have revealed this problem (as you do here), nor the observation-based source apportionment in Mahowald et al., 2008, which was consistent with the much smaller combustion sources. Instead one should say perhaps: Wang et al. (2014) taking into account the potential volatilized-P produced during combustion processes by assuming that all mismatches between observed deposition (<1000um aerosols) and modeled-long-range transported (<20um) deposition was due to combustion, estimated about 30 times higher global atmospheric P emissions from biomass burning and anthropogenic combustion processes (0.7 Tg-Pyr-1 and 1.8 Tg-P yr-1 respectively)." Or simply don't refer to that paper here or mention it in passing, since it is deeply methodologically flawed. [J Brahney et al., 2015] discusses how to compare to the Tipping et al., data in a more explicit aerosol size manner, and extends the Mahowald et al., 2008 study, showing that one can match deposition and concentration observations at the same time.**

Matching atmospheric deposition fluxes and concentrations at the same time is also what we try to do in the present study focusing on coastal and oceanic regions. See also our reply to reviewer's detailed comments point 3.

**5)        "where EDu is the on-line calculated dust emissions in the model, F880 is a factor applied to adjust the P emissions to the global mean P content of mineral dust in the model domain of 880 ppm per weight as observed by Zamora et al. (2013), and EP is the resulted inorganic P emissions from mineral. P-containing minerals associated with dust particles are emitted in the fine and coarse mode with mass median radii (lognormal standard deviation) of 0.34um (1.59) and 1.75um (2.00), respectively. The apatite emissions from mineral dust calculated for the year 2008 amount to 1.034 Tg-P yr$^{-1}$ with 10% of it (0.103 Tg-P yr$^{-1}$) in the dissolved form (Table 1)." How does this approach compare to the size resolved methods used in [J Perlwitz et al., 2015] for this mineral?**

Perlwitz et al. (2015) study focused on Fe-containing minerals. For the present study, we did not account for different P content in different dust minerals since that information was not available in the database that we have used or between soil and aerosols. Although the repetition of Perlwitz et al. (2015) methods for apatite minerals is out of the scope of this study, we added a comment in the manuscript in section 2.1.1:

"In a recent iron modelling study however (Perlwitz et al., 2015), a significant effort has been made to model the mineral composition of dust considering the differences from the original soil composition. Perlwitz et al. (2015) have found significant overestimate (a factor of 10-30) mainly in the fine aerosol emissions that are the smallest part of dust emissions (e.g. about 7% of the total emissions in our model) and an underestimate in the larger particles emissions both for total dust and for individual minerals when the mineralogy of dust aerosol is assumed to be the same as that of the soil. However for the present study, we did not account for different P content for dust particles in the fine and the coarse mode, since the global soil mineralogy dataset used (Nickovic et al., 2012) does not provide any information of P content in silt and clay soil particles separately."

**6)      Section 2.0: model description; can you describe your aerosol size bin or modal structure for the primary aerosols in P?**

We use modal scheme and this is clarified in the introduction of section 2.0, see also our detailed reply to the comment 8 of reviewer 1.

**7)      Section 2.1.3: how do you include bits of insects and plants that would be part of PBAP? How important is the neglect of these terms to your budgets do you think?**

PBAPs from insect fragments and plant debris are neglected in the present study. Omission of these super coarse particles is expected to lead to an underestimate in the PBAPs contribution to P deposition that requires to be evaluated with targeted observations. This is now clearly stated in the beginning of section 2.1.3.

We also added the following sentence in section 4.3: "Note that as mentioned in section 2, PBAPs from insect fragments and plant debris are neglected in the present study. Thus the contribution of PBAPs to bioavailable P deposition is here underestimated."

**8)      Please fix English by adding preposition (e.g. of): "(i.e. Nigeria downwind the Sahara Desert, Pakistan downwind the Thar Desert and China downwind of the Gobi desert)."**

Done.

**9)      In Fig. 4b, PO4 deposition fluxes (wet and dry deposition) from the Vet et al. (2014) compilation and from observations at Finokalia Station (Mihalopoulos and co-workers, unpublished data) are also compared with the model derived fluxes for the PRESENT simulation." What is the size distribution of the PO4 in the deposition? Is it the same size as the modeled boxes? I also think you should present the data you are going to compare against as a section in the methods, and describe the characteristics of the data, especially as some of the data is from unpublished sources. We also need to know where this data comes from physically: is it all in Greece, or elsewhere?**

A subsection 2.3 describing the data for model evaluation has been added in section 2, and the description of all data is now provided in the supplementary tables S1 and S2. In addition, Figures 4, S4, S5, S6 have been modified for clarify. Size segregated comparisons are now shown in these figures. Figures 4a and 4b illustrate the global distribution of the locations with aerosol concentrations and deposition fluxes data used for the model evaluation respectively.

**1.1      The subsection 2.3 follows: "Observation data for model evaluation**

The evaluation of the global atmospheric P cycle for the present study has been performed based on available observations of aerosol concentrations (Table S1) and deposition fluxes (Table S2) from various locations around the globe (cruises and land-based stations). The methodological details of the observations used for this study are well documented in the literature and thus are not reviewed here in detail. For DP concentrations in ambient aerosols, we compiled cruise observations of $PO_4^{3-}$ over the Atlantic Ocean (50°N–50°S) from Baker et al. (2010), over the Western Pacific (25°N–20°S) from Martino et al. (2014) and over the Eastern Tropical North Atlantic Ocean (58°S–35°N, 14°–38°W)

from Powell et al. (2015). For these oceanic cruise observations, samples were either collected separating into fine- (aerodynamic particle diameter < 1µm) and coarse-mode (1µm< aerodynamic particle diameter) particles using cascade impactors that may include or exclude particles with diameters larger than 10 µm, or using a single bulk filter. We additionally use average $PO_4^{3-}$ concentrations (aerodynamic particle diameter < 10µm) from cruise measurements over Bay of Bengal and the Arabian Sea (Srinivas and Sarin, 2012). Finally, we also took into account land-based TP and $PO_4^{3-}$ aerosol concentrations measurements from two sites in the Mediterranean i) from the Finokalia monitoring station (35°20`N, 25°40`E) located in the Eastern Mediterranean (Crete, Greece) and ii) from Ostriconi (42°40`N, 09°04`E) located in the Western Mediterranean (Corsica, France). The samples at both sites were collected either separating for the fine- (aerodynamic particle diameter < 1.3 µm) and the coarse-mode (10 µm > aerodynamic particle diameter > 1.3 um) (Koulouri et al., 2008; Mihalopoulos and co-workers, unpublished data) or as bulk (Markaki et al., 2010). Details about the characteristics of these Mediterranean sampling sites can be found in Markaki et al. (2010), while the methodology for aerosol sampling and analysis is described in detail in Koulouri et al. (2008).

Although P deposition fluxes data are rather limited on a global scale, for the present study we use the wet and dry deposition fluxes (both for TP and DP) compiled by Vet et al. (2014) (R. Vet, personal communication, 2016). For wet deposition of DP, we use available filtered (i.e. analyzed as orthophosphates with no digestion as DIP) and unfiltered (i.e. analyzed as orthophosphates following digestion as total DP) annual measurements (Fig. 8.2 in Vet et al., 2014). For the TP wet deposition measurements we use annual wet deposition measurements (Fig. 8.3 in Vet et al., 2014) of unfiltered samples. The compilation of the phosphorus dry deposition fluxes by Vet et al. (2014) is based on airborne phosphorus (TP and PO4) concentrations from around the world and gridded annual dry deposition velocities from the Mahowald et al. (2008) modelling study (Fig. 8.6 and Fig. 8.7 in Vet et al., 2014). The size distribution used in these dry deposition calculations, is the same as in the modelling study by Mahowald et al. (2008), thus the derived dry deposition fluxes account for particles with diameter up to 10 µm. Finally, we also take into account DP wet and dry deposition observations from the Finokalia Station in the Eastern Mediterranean (Markaki et al., 2010; Mihalopoulos and co-workers, unpublished data), based on rain water samplings (wet only collector) and glass-bead devices respectively. Further details on the methodology of the deposition measurements at Finokalia can be found in Markaki et al. (2010)."

**10)    "(MNB; see definitions of statistical parameters in Myriokefalitakis et al. (2015))" You also need to describe your methods in the methods section: it is not ok to refer us for basic information to another paper.**

We have now included the definitions of this statistical parameter – mean normalized bias (MNB) in section 2.3.

**11)    Figure 7: maybe you want to reformat so that there won't be too much white space in the final figure for this?**

We have reformatted the figure as suggested.

**12)    The present-day P solubility of deposited aerosols (hereafter SP = %DP/TP) is calculated to vary spatially significantly (Fig. 7a)," vary spatially significantly is awkward: please rephrase and only use significantly if you mean statistically significantly.**

We rephrased by removing 'significantly'

**13)    For your past and future estimates: Your P is strongly dependent on dust, and yet you don't include any changes in dust. I don't think you need to add much here, but just some statements that dust appears to vary strongly and perhaps be sensitive to humans climate change and/or land use [P Ginoux et al., 2012; N Mahowald et al., 2010; J Prospero and P Lamb, 2003], and thus that would also be an important drivers of changes in P and SP.**

We agree with the reviewer that past and future dust sources may be changed due to global change. In our model, P atmospheric cycle is strongly depended on dust outbreaks, since according to our calculations about 50% of the deposited bioavailable P is originated from soils for the present atmosphere. As recommended we added the following sentence in section 4.1. of the manuscript: "
[revised manuscript text omitted]

---

## Author Comment (AC3) · 31 Oct 2016

Dear Editor,

We would like to thank the referees as well as Dr. Bikkina Srinivas for their comments that helped improving the presentation of our study.

We have addressed all their concerns as it is detailed in the point-by-point replies to the referee's comments that we have posted in the open discussion session.

In particular, following the referee #1 comments, we have performed a new 11-year interannual global simulation 2000-2010 in 3ox2o (lon x lat) horizontal resolution. All figures and results have been updated accordingly. The comparisons to observations

are now not only day and location specific but also year specific as requested by the referee. As expected, no changes of importance were seen in the results and the model evaluation.

We have also performed again the sensitivity simulation using Wang et al (2014) anthropogenic emissions of P, following the referee' suggestion. Our conclusions remained unchanged. The presentation of our methodology has been improved for clarity and to avoid misunderstandings. We have also added section 2.3 in which we present the compilation of observations that are used for evaluation of our model results and which are detailed in the supplementary material Tables S1 and S2. In section 3.5 on solubility we have summarised available observations that provide hints on P solubility in the atmosphere and added a thorough discussion on the limitation when comparing these observations to our simulations (see details in our reply to referee #2 major comment 4).

Finally, the section 4.2 with the discussion on implications of our results for the biogeochemical cycles and on uncertainties as well as the conclusions have been further developed following both referee's comments.

We hope that our manuscript is now suitable for publication in Biogeoscience.

With kind regards,

The authors